

# Model simulations of arctic biogeochemistry and permafrost extent are highly sensitive to the implemented snow scheme

Alexandra Pongracz[1], David Wårlind[1], Paul A. Miller[1], and Frans-Jan W. Parmentier[1,2]

[1]Department of Physical Geography and Ecosystem Science, Lund University, Lund, Sweden.
[2]Centre for Biogeochemistry in the Anthropocene, Department of Geosciences, University of Oslo, Oslo, Norway

**Correspondence:** Alexandra Pongracz (alexandra.pongracz@nateko.lu.se)

**Abstract.** The Arctic is warming rapidly, especially in winter, which is causing large-scale reductions in snow cover. Snow is one of the main controls on soil thermodynamics, and changes in its thickness and extent affect both permafrost thaw and soil biogeochemistry. Since soil respiration during the cold season potentially offsets carbon uptake during the growing season, it is essential to achieve a realistic simulation of the effect of snow cover on soil conditions to more accurately project the direction

of arctic carbon-climate feedbacks under continued winter warming.

The Lund-Potsdam-Jena General Ecosystem Simulator (LPJ-GUESS) dynamic vegetation model has used – up until now – a single layer snow scheme, which underestimated the insulation effect of snow, leading to a cold bias in soil temperature. To address this shortcoming, we developed and integrated a dynamic, multi-layer snow scheme in LPJ-GUESS. The new snow scheme performs well in simulating the insulation of snow at hundreds of locations across Russia compared to observations. We

show that improving this single physical factor enhanced simulations of permafrost extent compared to an advanced permafrost product. Besides soil thermodynamics, the new snow scheme resulted in increased winter respiration and an overall lower soil carbon content due to warmer soil conditions. The *Dynamic* scheme also influenced vegetation dynamics, resulting in an improved vegetation distribution and tundra-taiga boundary simulation.

This study highlights the importance of a correct representation of snow in ecosystem models to project biogeochemical

processes that govern climate feedbacks. The new dynamic snow scheme is an essential improvement in the simulation of cold season processes, which reduces the uncertainty of model projections. These developments contribute to a better understanding of the Arctic's role in the global climate system.

## 1 Introduction

The Arctic is undergoing rapid warming, with some of the most pronounced changes occurring during the winter (Box et al.,

2019; Natali et al., 2019). As a result, snow thickness, the extent of snow covered area, and snow season length are decreasing, and this is expected to continue in the future (AMAP, 2017; IPCC, 2014). Snow is an important abiotic component of the Arctic system, since it provides an insulating cover for vegetation and soil. Snow insulation is recognised as the primary control over soil thermodynamics (Lawrence and Slater, 2010b), and soil temperature is closely connected to physical (i.e. permafrost active layer depth) and biogeochemical (i.e. decomposition, greenhouse gas emission) processes (Peng et al., 2016). Observa-





tions show that snow cover changes have played a major role in a warming trend of permafrost soils of approximately 0.3 °C
per decade (Biskaborn et al., 2019; AMAP, 2017). This warming may lead to increased microbial activity, decomposition rates
and bioavailability of previously frozen soil carbon. Since permafrost soils contain approximately 1600 Pg carbon, accounting
for half of the global soil carbon storage (Hugelius et al., 2014), there is ample potential for these changes to lead to a release
of the greenhouse gases $CO_2$ and methane. This has the potential to accelerate global warming (Schuur et al., 2015), which

underlines the need for a better understanding of drivers and potential feedbacks to better predict the rate and magnitude of
future carbon exchange.

Despite numerous field-based and modelling efforts to date, it is still uncertain whether the Arctic will act as a carbon source
or sink in the future (McGuire et al., 2012; Virkkala et al., 2021). The predicted future carbon balance varies widely among

models depending on the representation and level of detail of key processes such as soil temperature and vegetation dynamics
(Schuur et al., 2015; McGuire et al., 2018). One of the key goals of model development is to decrease uncertainty of simula-
tions by refining these processes. While extensive research has been carried out on the mechanics of the growing season, few
studies have been directed at cold season processes. Recent studies suggest that the contribution of the non-growing season to
the annual carbon budget may have been underestimated (Pirk et al., 2015; Mastepanov et al., 2013). A recent meta-analysis

by Natali et al. (2019) found significant wintertime carbon loss, and highlighted the large spread in model simulations of non-
growing season greenhouse gas emissions. Models generally underestimated the observed winter flux emissions due to the
inaccuracies in their simulation of cold season respiration. Natali et al. (2019) stress the need to revise the impact of environ-
mental drivers and feedbacks in models. Collectively, these efforts demonstrate that the influence of cold season processes on
the annual carbon balance is larger than previously suggested.

The ability of models to simulate physical and biological processes in the soil is limited by the complexity of their representa-
tion of snow. A recent snow-related model evaluation project analysed the performance of models with different complexity –
focusing on variables such as snow-covered area and snow season length. This SNOWMIP found that a dynamic simulation of
internal snowpack processes, such as density and temperature calculations, is critical to simulate snow thermal profiles (Krin-
ner et al., 2018). In addition, more complex snow schemes perform better when simulating cold season processes (Vionnet

et al., 2012; Slater et al., 2017; Wang et al., 2016). To balance computational efficiency and the need for detail, most ecosystem
models use an intermediate complexity, multi-layer snow module (Vionnet et al., 2012; Krinner et al., 2018). Such schemes
may not capture fine-scale internal snowpack processes such as the evolution of high-density wind slab layers, but they are
complex enough to simulate key physical processes that influence the thermal dampening property of snow. Since LPJ-GUESS
had a single-layer, static snow representation, it was found to deviate from observational records of air-soil temperature rela-

tionships – simulating cooler winter conditions and performing poorly when compared to 8 land surface models (Wang et al.,
2016). This showed, combined with previous research, that the snow representation in LPJ-GUESS needed to be revised to
better capture Arctic cold season conditions.





The primary aim of this study is to improve the capacity of LPJ-GUESS to simulate the insulating capacity of snow. By
developing and integrating a dynamic, intermediate complexity snow scheme, we also aim to improve the soil temperature
and biogeochemistry simulation. To investigate this effect of the new snow scheme in LPJ-GUESS, we set out to quantify
the impact on physical variables, i.e. the direct impact of snow insulation on soil temperature and permafrost conditions. To
further evaluate the snow-related influence on biogeochemistry - such as changes in growing season length - we analyse a set
of biogeochemical variables. Due to differences in soil thermodynamics, we expect to see changes in ecosystem productivity,
heterotrophic respiration and soil carbon pools. Moreover, we address the changes to vegetation dynamics and composition.
The updates to the model will allow us in the future to assess other snow-related processes and feedbacks on a global scale,
such as the impact on surface albedo and food access to herbivores.

## 2  Materials and methods

LPJ-GUESS is a process-based dynamic vegetation model, widely applied on a regional and global scale (Smith et al.,
2001, 2014). For this study, we used a customised Arctic version of LPJ-GUESS 4.0 (subversion 8899). The model simu-
lates soil freeze-thaw processes and is applicable to studies of processes at northern high latitudes (Miller and Smith, 2012). In
this study we restrict simulations to the northern circumpolar region, above 60° latitude, with a spatial resolution of 0.5°x 0.5°.
The CRUNCEP global reanalysis climate dataset version 7 was used as input for all of our model simulations (Viovy, 2016).
We ran the model with a 500 year spin-up period to establish an equilibrium vegetation state and a 40 000 year offline spin-up
period for soil conditions.

LPJ-GUESS simulates vegetation dynamics on an individual and patch scale, taking into account growth, competition for re-
sources and disturbances. This feature makes it possible to assess how changes in environmental conditions affect vegetation
distribution and composition. In this study, we applied 15 plant functional types (PFT) that characterise Arctic ecosystems
(see Table A1 in the Appendix). For more details on the model structure, see Smith et al. (2001, 2014); Wania et al. (2009),
and references therein. To assess the newly developed, intermediate complexity snow scheme's performance and influence, we
conducted simulations with both the old *Static* and the new *Dynamic* snow schemes.

### 2.1  *Static* snow scheme

The *Static* snow scheme, which has been in use in LPJ-GUESS until now, treats snow as a single layer with constant values for
thermodynamic parameters. Snowfall is simulated on any given day when precipitation ($P$, $mm$) occurs and air temperature
($T_k$, °C) is at or below $T_{max}$ (°C) – which is the temperature maximum at which precipitation occurs in snow form. $T_k$
denotes the temperature of a layer $k$, in this case the air layer. Snow density ($\rho_k$, $kg\ m^{-2}$) and snow thermal conductivity
($K_k$, $Jm^{-1}K^{-1}s^{-1}$) are constant at 362 and 0.196, respectively. Snow heat capacity ($C_k$, $Jm^{-3}K^{-1}$) is calculated by Eq.
(1) (Fukusako, 1990).

$$C_k = 1000\rho_k(0.185 + 0.00689T_k) \tag{1}$$





Compaction processes are not represented in the *Static* scheme. Snowmelt ($melt$, $mm$) is governed by air temperature and follows a linear function as shown in Eq. (2), (Choudhury and DiGirolamo, 1998).

$$melt = (1.5 + 0.007 \times P)(T_k - T_{max}) \tag{2}$$

The snowpack is homogeneous in its physical properties, and neither internal processes nor seasonal dynamics are simulated using the *Static* scheme. Using this approach, snow conditions are assumed to be uniform across the Arctic regardless of air temperature regime or seasonal snow dynamics.

### 2.2 *Dynamic* snow scheme

The schematic structure of the multi-layer snow scheme is shown in Figure 1. The occurrence of snowfall on any given day depends on air temperature and precipitation, using the same principle as for the *Static* scheme. Fresh snow density ($\rho_{fresh}$, $kg\ m^{-3}$) is calculated by taking into account air temperature and wind speed, following Eq. (3), where $a$, $b$ and $c$ are scaling parameters (for parameter values see Table A2 in the Appendix).

$$\rho_{fresh} = a + b \times T_{max} + cU_{10}^{0.5} \tag{3}$$

$U_{10}$ denotes the 10 m height wind speed ($m\ s^{-1}$), following the detailed snowpack model Crocus (Vionnet et al., 2012). To avoid unrealistically low snow density values, the density minimum is set to 100 $kg\ m^{-3}$.

The new snow scheme simulates internal snowpack dynamics with up to five snow layers, taking into consideration each layer's depth. Fresh snow either initiates a snowpack or is added to already existing snow layers. If the freshly fallen snow is added to the snowpack, the physical properties of the snow layer are updated. The number and thickness of snow layers are defined according to predefined thresholds: a new snow layer is initialised when an existing layer exceeds twice the prescribed threshold height (2 x 100 mm). If a single snow layer exists, but does not reach the minimum height (set to 50 mm), the shallow snow layer properties are combined with the top soil layer. Thereafter, their properties are scaled using weighted averages based on the layer's ice, water and air fractions for the sake of computational stability. In case all five layers are exceed the prescribed maximum threshold, the bottom layer accumulates snow in order to preserve and align vertical resolution near the surface of the snowpack. The snow layer density ($\rho_k$) and depth ($z_k$, $m$) relationship is described by Eq. (4), where $I_k$ ($kg\ m^{-2}$) defines the ice content of a layer (Lawrence et al., 2019)).

$$z_k = \frac{I_k}{\rho_k} \tag{4}$$





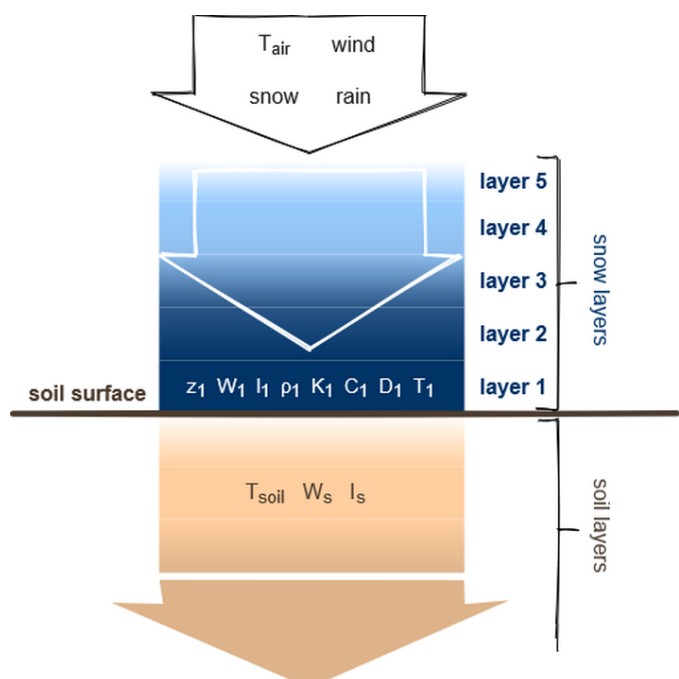

**Figure 1.** Snow pack structure and physical properties. Where $z$ shows layer depth, $W$ water content, $I$ ice content, $\rho$ density, $C$ heat capacity, $K$ thermal conductivity, $D$ thermal diffusivity and $T$ shows layer temperature. A detailed list of the used variables can be found in Table A2.

The density of a snow layer changes through compaction, which is simulated by two processes: (1) mechanical compaction due to pressure from the overlying snow layers as shown in Eq. (5) Best et al. (2011).

$$\frac{\partial \rho_k}{\partial t} = \frac{\rho_k g M_k}{\eta_k} \exp\left(\frac{k_s}{T_{max}} - \frac{k_s}{T_k} - \frac{\rho_k}{\rho_0}\right) \tag{5}$$

The increase in the snow layer's density ($\partial \rho_k$) depends on the mass of overlying layers ($M_k$, $kg$). $\eta_k$ ($10^6 Pa\ s$) denotes the compactive viscosity factor, while $k_s$ is an empirical constant defined by Best et al. (2011) with a value of 4000 K. Snow

density may also change by (2) phase changes as a result of freeze-thaw processes within the layers. If a layer's snow or liquid water content changed during freeze-thaw events, its depth and density properties are recalculated, taking into account the snow and ice fractions of the layer as shown in Eq. (4).

In contrast to the *Static* formulation, phase changes within the snow layers depend on the layer's internal temperature, and this controls the melting process in the *Dynamic* snow scheme. This development enables the simulation of mid-winter melt events

and ensures an improved representation of internal snowpack thermodynamics. Upon melt, each layer can retain a fraction of





liquid water based on Eq. (6), where $rw_{min}$, $rw_{max}$ are empirical constants and $\rho_t$ is a reference density (Wang et al., 2013; Anderson, 1976).

$$W_{cap,max} = I_k[rw_{min} + (rw_{max} - rw_{min}) \times max(0, \frac{(\rho_t - \rho_k)}{\rho_t})] \tag{6}$$

If the liquid water content ($W_k$, $mm$) of a layer exceeds the maximum water holding capacity ($W_{cap,max}$, $mm$), water is
passed to the layer underneath following a simple bucket model. Rain on snow events (ROS) are simulated if it rains while a snowpack is present. The energy of rainwater may induce phase changes in the snow layers. The overflow liquid water is forwarded to the underlying snow layers and lastly to the top soil layer to percolate to the soil or to be discharged as runoff. Each layer is characterised thermodynamically by the following physical properties: density, temperature, thermal conductivity, heat capacity and diffusivity ($D_k$, $m^2$ $day^{-1}$). Thermal conductivity is calculated using density as shown in Eq. (7) (Best et al.,
2011), following a power function ($Jm^{-3}K^{-1}$).

$$K_k = 2.22 \left(\frac{\rho_k}{\rho_0}\right)^{1.88} \tag{7}$$

Heat capacity is determined by taking into account snow layer density and temperature according to Eq. (1). The snow diffusivity is calculated by Eq. (8). Soil and snow layer temperatures are computed, taking into account each layer's thermal conductivity, heat capacity and height, using the Crank-Nicholson finite difference method to solve Eq. (9) (Lawrence et al.,
140 2019).

$$D_k = \frac{K_k}{C_k} \tag{8}$$

$$\frac{\partial T}{\partial t} = \frac{\partial}{\partial z}\left(D(k)\frac{\partial T}{\partial z}\right) \tag{9}$$

The computational cycle ends by rearranging the layers based on the depth thresholds, taking into account the potential liquid water content. Meltwater is passed to the soil for percolation after this step. This cycle is repeated each day when there is a
snow or rain-on-snow event. The daily snow cycle of the *Dynamic* snow scheme is depicted in Figure 2.




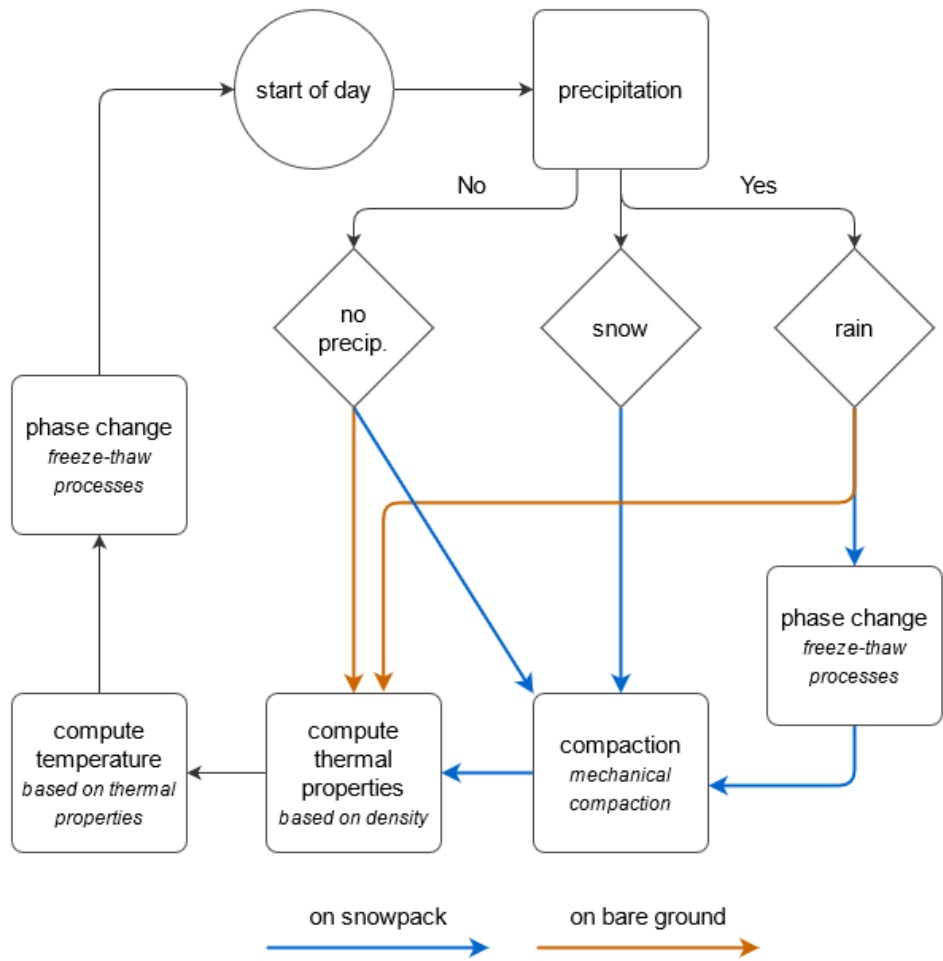

**Figure 2.** Steps in the daily computational cycle for the *Dynamic* snow scheme. Blue arrows indicate workflow in case snow exists on the ground, while orange arrows show following steps on bare ground.

Besides the changes in the representation of snow, the calculation of heterotrophic respiration below 0 °C was changed following a recent data synthesis (Natali et al., 2019) to better represent arctic conditions. This adjustment was implemented for both the *Static* and *Dynamic* schemes. The minimum decomposition temperature was set to -20 °C and the Q10 value to 2.9. A comparison between the old and new functions is shown in the supplement (Fig. S1). This adjustment led to higher soil
respiration in both schemes during the cold season compared to the old model set-up.

The implemented processes and physical representations are simpler than in dedicated, high-resolution snow models – such as Crocus and SNOWPACK (Lehning et al., 2002; Vionnet et al., 2012) – but reflect the model improvements identified as being most important in previous model inter-comparison studies (Krinner et al., 2018). These improvements enable us to simulate a
more realistic range of snow conditions and soil thermal conditions across the Arctic.





## 2.3 Simulation set-up

The performance of LPJ-GUESS using both snow schemes was compared at a site and regional level. Modelled properties were compared to observational datasets, when available. We quantified the correspondence between simulated and measured variables using statistical methods. A detailed single site model-data comparison of the internal structure of the snowpack can be found in the supplement (Fig. S2).

i) Site-level comparison

We ran the model for five well-studied northern high latitude sites in order to identify how well the two snow schemes can simulate snow and soil temperature at a site-level. These sites are Abisko, Zackenberg, Bayelva, Kytalyk and Samoylov – see site details in Table S1 in the supplement. Measurements of snow depth and soil temperature were sorted and averaged on a daily basis – 10 years for the simulations, and all available years for observations at each site. Model outputs were examined and compared to these time-series to evaluate the snow schemes' ability to simulate snow depth and soil temperature seasonality adequately.

ii) Regional simulations

We conducted simulations for a set of Russian sites which were part of the study by Wang et al. (2016), as a follow-up, and re-evaluate the snow insulation effect in LPJ-GUESS over a large region. First, snow depth and soil temperature data were sorted monthly for each site (n=256) for the years 1980-2000. Site observations were provided by the All-Russian Research Institute of Hydrometeorological Information – World Data Centre (RIHMI-WDC; http://meteo.ru/). Following this, averages were calculated for December, January and February. The difference between soil (25 cm depth) and air temperature - henceforth $\Delta T$ - was used as a proxy to evaluate the strength of the model-simulated insulation effect. Snow depth, soil temperature and $\Delta T$ series were grouped according to air temperature to evaluate the insulation capacity under different temperature regimes.

iii) Pan-Arctic simulations

Finally, we conducted model simulations across the Arctic to assess the influence of snow on selected physical and biogeochemical variables and vegetation properties. When applicable, variables were averaged over December, January and February to emphasise the effect on the winter season. Instead of the absolute results, we show the difference between the set of simulations, calculated as the difference between *Dynamic* and *Static* model outputs.

## 3 Results

### 3.1 Site level simulations

To assess the performance of the two snow schemes we composed seasonal cycles based on monthly averages of near-surface soil temperature at 25 cm depth (a) and snow depth (b), shown in Fig. 3. The corresponding root mean squared error (RMSE) for each study site is shown in Table 1.





Generally, the *Dynamic* scheme shows only minor improvements in the simulation of snow depth. Despite this, modelled soil temperatures are much closer to the observed values for all sites, especially during the winter months. This behaviour highlights that changing the internal snowpack dynamics with the *Dynamic* snow scheme had a significant effect on soil thermodynamics,

190 even when the simulated snow depth differed marginally. There is a deviation between modelled and measured soil temperature in the spring and summer seasons, where the simulated soil temperature is higher than the observations. This phenomenon is addressed and discussed in the following sections. The statistical comparison (*site statistics*) shows that there is a smaller variance of modelled values of snow depth and soil temperature using the *Dynamic* snow scheme, which indicates an improvement in comparison to the *Static* simulations' outputs. The RMSE (Table 1) also shows that the *Dynamic* scheme provides an

195 improved fit of simulated soil temperature and snow depth at most sites. Overall, we conclude that, with the *Dynamic* scheme, the model is able to simulate snow and soil temperatures that correspond better with the observed ranges.

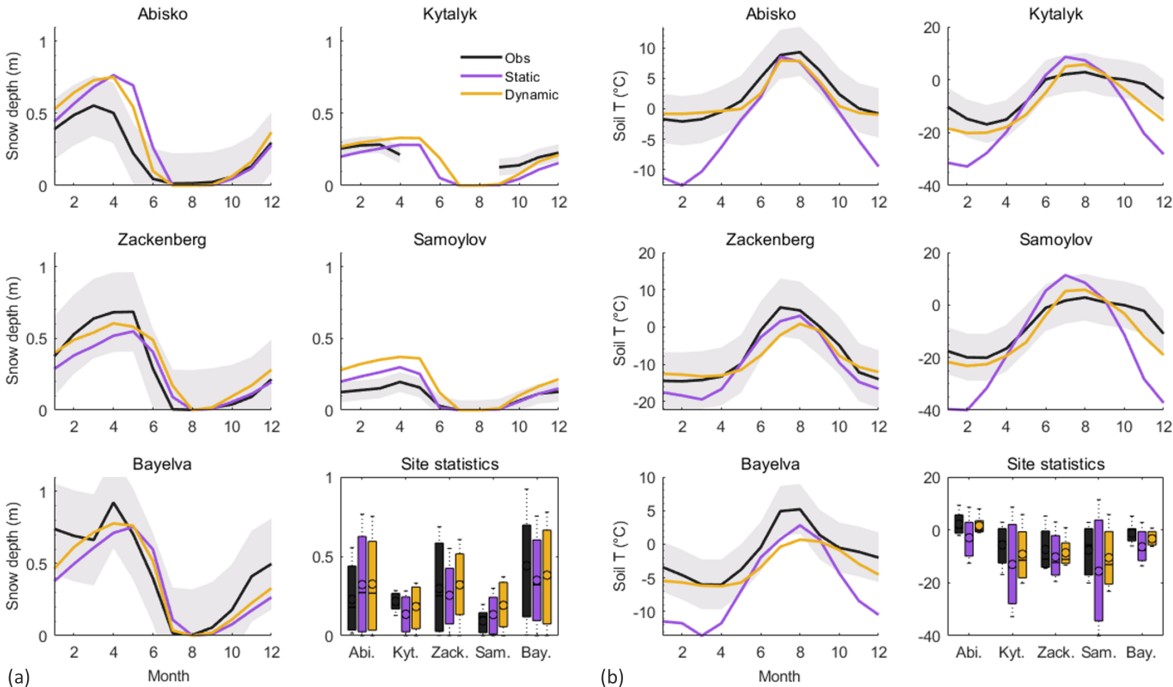

**Figure 3.** Seasonal cycle of (a) soil temperature at 25 cm depth and (b) snow depth for the studied sites, comparing model simulations and observations. The shaded grey area shows the 95% confidence interval around observations. Site statistics show the spread of monthly snow depth and soil temperature values for the respective sites.



**Table 1.** RMSE for soil temperature and snow depth for the applied snow schemes for the single site simulations.

|  | snow scheme | Abisko | Kytalyk | Zackenberg | Samoylov | Bayelva |
|---|---|---|---|---|---|---|
| Soil T (°C) | *Static* | 6.13 | 12.35 | 3.17 | 14.97 | 5.65 |
|  | *Dynamic* | 1.38 | 5.23 | 3.39 | 4.73 | 2.53 |
| Snow depth (m) | *Static* | 0.17 | 0.09 | 0.11 | 0.06 | 0.18 |
|  | *Dynamic* | 0.14 | 0.07 | 0.10 | 0.13 | 0.13 |

## 3.2 Russian site simulations

Following the *Dynamic* scheme's improved performance at the site level, we further evaluate the model's performance at the regional scale for the same sites as in the previous model intercomparison by Wang et al. (2016) that highlighted shortcomings in the snow scheme of LPJ-GUESS. Figure 4 shows the snow insulation effect over a set of Russian sites, using the two snow schemes, where the coloured bars show different temperature regimes. The figure is compiled from 20 winter season average values of near-surface soil temperature (25 cm depth) and snow depth per site. Due to the air temperature-based classification, the number of samples per bin is not balanced, which led to an uneven number of values allocated to the different groups. The top row of Fig. 4 shows that the *Dynamic* snow scheme has a better skill in simulating the relationship between soil temperature and snow depth than the *Static* scheme. It must be noted that there is a clear difference between the current *Static* scheme simulations and results reported by Wang et al. (2016), which is due to recent updates in the model, independent of the snow module, and the different climate forcing dataset used in this study.

It is apparent from the $\Delta T$ and snow depth relationship (Fig. 4, bottom row) that the *Dynamic* scheme reproduces the observed insulation effect well. Unlike the *Static* scheme, the new snow module can also simulate the different insulation behaviour depending on the air temperature regimes. The improved performance of the *Dynamic* scheme is confirmed by the root mean squared error (RMSE), shown in the supplement (Table S2). RMSE decreased significantly both for the soil temperature-snow depth and $\Delta T$-snow depth relationships. This regional analysis confirmed that the new *Dynamic* snow scheme has an improved skill to simulate winter soil conditions.





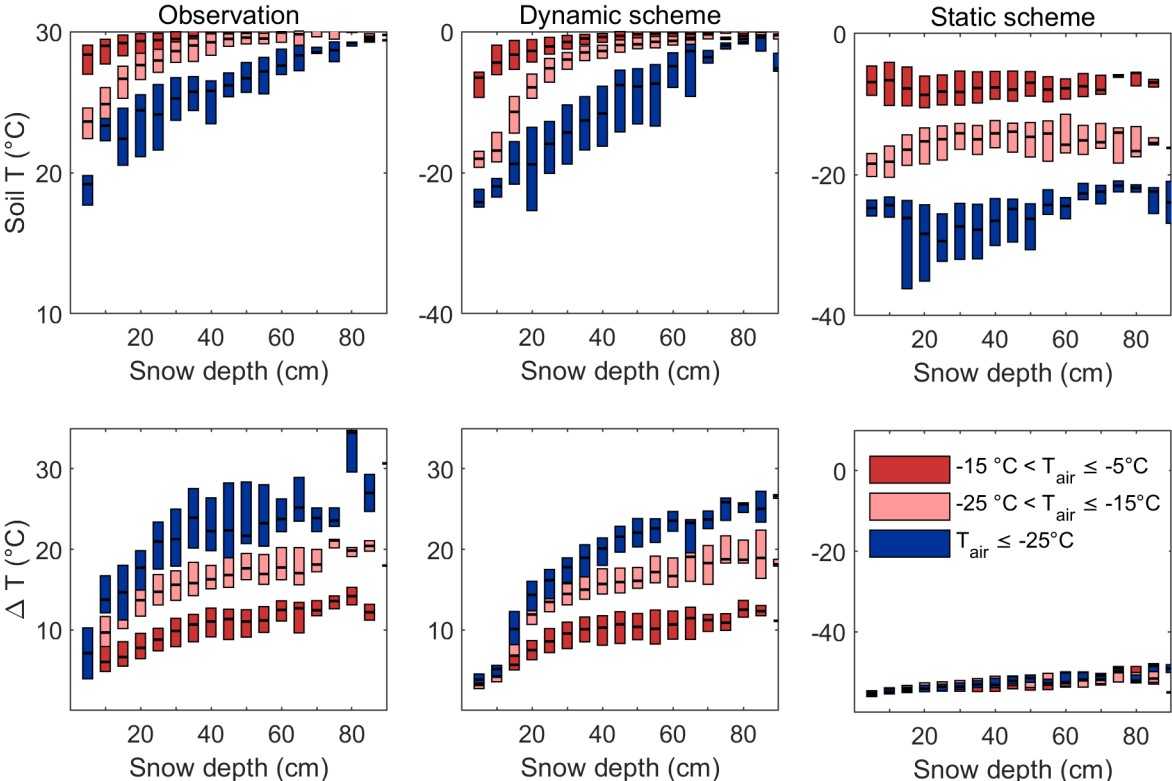

**Figure 4.** Comparison of the observed and modelled snow insulation effect at the Russian sites between observations and model simulations using the *Dynamic* and *Static* schemes. Top row: soil temperature and snow depth relationship. Bottom row: difference in air-soil temperature and snow depth relationship. Snow depth presented on the horizontal axis is classified in 5 cm depth bins. Colours indicate different air temperature regimes and upper and lower bars the $25^{th}$ and $75^{th}$ percentiles.

## 3.3 Pan-Arctic simulations

### 3.3.1 Impacts on physical variables

To assess how the two snow schemes differ in simulating seasonal snow across the Arctic, we subtracted output variables from simulations with the *Static* module from those with the *Dynamic* module. We calculated average conditions for winter
(Dec-Jan-Feb) and summer (Jun-Jul-Aug) for the period 1990-2015. Figure 5 (a) shows the difference in simulated wintertime snow depth. The *Dynamic* scheme shows an overall higher snow depth across the Arctic with the most pronounced changes in coastal Scandinavia and in Western-Russia. On average, the snow depth for the *Dynamic* scheme is 6 centimetres higher due to the implementation of compaction processes. The mean pan-arctic seasonal dynamics of snow depth, soil temperature and upper soil water content are shown in the supplement.





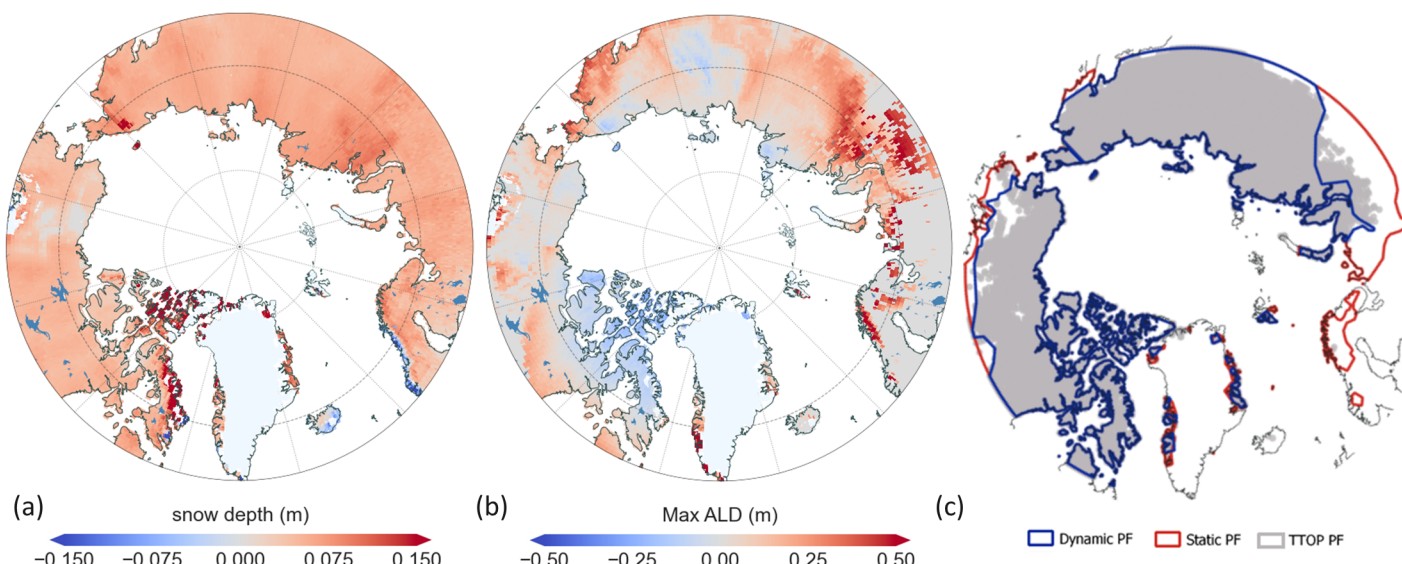

**Figure 5.** (a) Snow depth difference in winter months and (b) maximum ALD difference, calculated by subtracting the *Static* from *Dynamic* simulation outputs. Modelled permafrost extent is based on mean annual ground temperature (MAGT), and plotted against the permafrost cover estimate by Obu et al. (2019) (TTOP model).

The main aim of developing the new snow scheme was not only to enhance the simulation of snow depth, but also to improve the simulation of snowpack properties that directly affect soil conditions. Therefore, we investigated how the internal structural changes in the representation of snow influenced soil thermodynamics. The soil temperature differences shown in Fig. 6 reveal that the new snow scheme influenced the winter season to a large degree, both within and outside of the permafrost region. Winter soil temperatures are higher with the *Dynamic* scheme, while it results in a cooler near-surface soil temperature during

the summer. A closer look at the monthly soil temperature values (supplement, Fig. S3 (a)) showed that May and June are cooler for the *Dynamic* scheme, but that the difference between the two schemes decreases towards the end of July. This underlines the hypothesis that the *Static* snow scheme has too little insulation and results in soil temperatures that are too cold during the winter months. Moreover, the *Static* scheme also does not insulate soils sufficiently during the springtime when air temperatures rise above 0 °C, which allows the soil to warm up more quickly even in the presence of a snowpack.



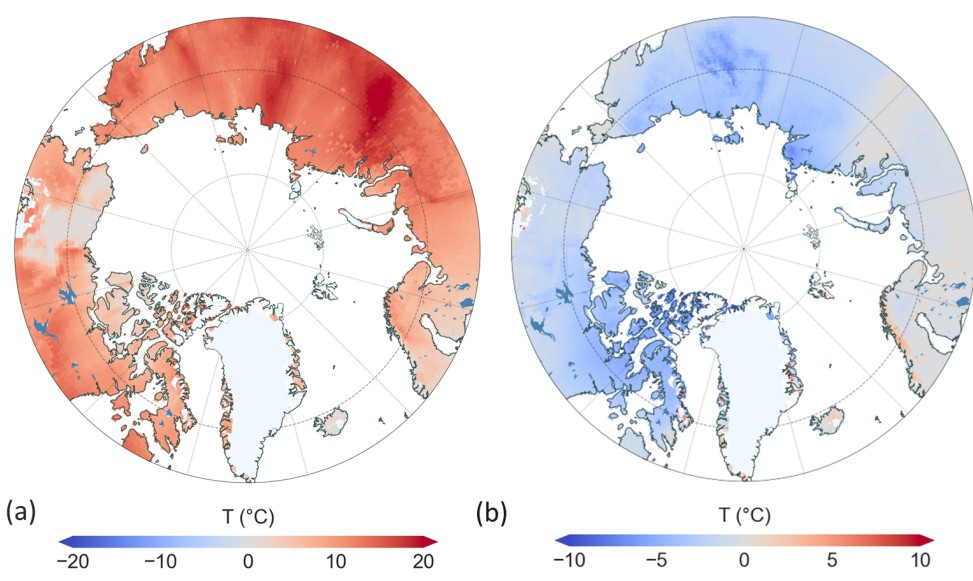

**Figure 6.** Near surface soil temperature (25 cm depth) difference between the *Dynamic* and *Static* simulations, for winter (a) and summer (b) seasons. Differences are calculated by subtracting the *Static* from *Dynamic* simulation outputs.

The depth to which the top soil thaws during summer, and refreezes in winter, in permafrost areas is called the active layer depth (ALD). The difference in the seasonal maximum active layer depth for the model simulations is shown in Fig. 5 (b). Since the *Dynamic* scheme had warmer soil temperatures, the modelled permafrost extent is smaller than with the *Static* scheme. We compared our model simulations with a recent satellite-driven permafrost extent estimate by Obu et al. (2019) – from here on referred to as the TTOP model. Modelled permafrost extent was defined by the area where the mean annual ground tempera-

ture (25 cm depth) was below zero. The *Dynamic* scheme's permafrost extent is close to the TTOP model's estimate, while the *Static* scheme simulates a much larger permafrost extent, as shown in Fig. 5 (c).

Besides governing the physical state of permafrost, snow and soil temperature also have a large influence on the temporal and spatial patterns of soil water content. Water availability is a key driver of the start of the growing season, nutrient avail-

ability and vegetation dynamics. The time-series analysis of upper soil water content highlights that the snowmelt rate is not significantly different between the schemes. Still, there is a large difference in soil temperature dynamics. Soil temperature for the *Static* scheme increases rapidly upon snow melt (supplement, Fig. S3 (a)), which results in an earlier onset of soil water availability and an increase in nitrogen mineralisation. This affects productivity, which we assess in the coming sections. Although the difference in water content and nitrogen mineralisation between the snow schemes converges towards zero as

summer progresses, we show that the change in snow scheme had a lasting effect beyond the cold season.



Overall, the new snow scheme had a substantial effect on winter soil thermodynamics. Due to differences in soil temperature dynamics, summer conditions were also altered by the snow scheme updates. It is apparent that the largest changes in snow depth, temperature and maximum ALD coincide. For instance, along the Scandinavian coast and Central Russia. Taken

together, our results show that the *Dynamic* snow scheme improved the simulation of physical variables.

### 3.3.2   Impacts on biogeochemical variables

Besides the impact on soil thermodynamics, we investigated how key biogeochemical components – such as productivity and carbon pools – were affected. The changes across seasons and permafrost conditions are summarised in Fig. 12.

Our simulated soil carbon pools deviate from literature values (Hugelius et al., 2014), and are consistently lower across the

Arctic. The main reason for this is the model's representation of soil organic matter processes. Soil carbon and nitrogen are represented by pools that exists in the top 50 cm of the soil column (Smith et al., 2014), and are thus only influenced by near-surface conditions. Moreover, peatlands are not explicitly represented. The differences in soil carbon between the schemes, as shown in Fig. 7 (a), coincide spatially with the highest differences in soil temperature. This suggest that the changes in soil temperature influence soil carbon in the model and therefore the potential for greenhouse gas emissions from soils. Vegetation

carbon pools (Fig. 7 (b), on the other hand, do not differ as much between the two snow schemes. Since the evaluation of soil carbon is not the focus of this study, soil carbon outputs were used to normalise the heterotrophic respiration to be able to interpret the relative differences between schemes (Fig. 8 (a) and (b)). With the *Dynamic* scheme, summer soil respiration decreased across the Arctic. Winter respiration, on the other hand, increased, except for Eastern-Russia. These changes in soil respiration can be attributed to changes in soil temperature, as shown in Figure 6.

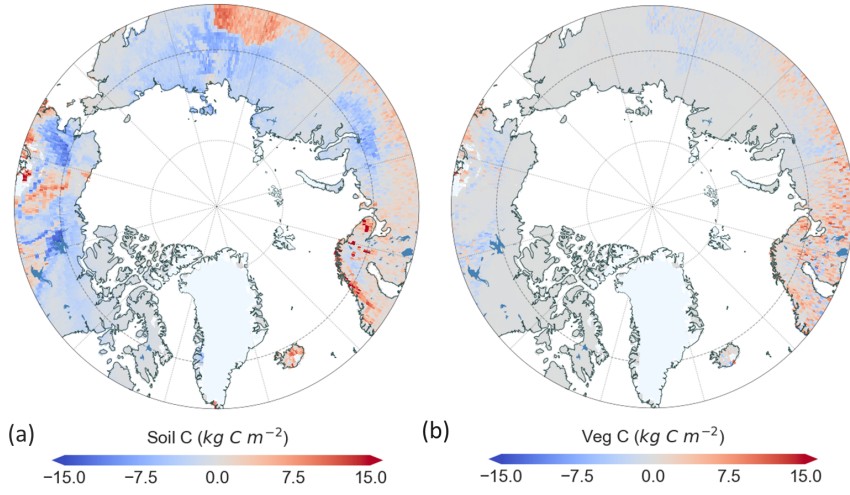

**Figure 7.** The difference in simulated soil (a) and vegetation carbon pools (b) between the two schemes. Differences are calculated by subtracting the *Static* from *Dynamic* simulation outputs.

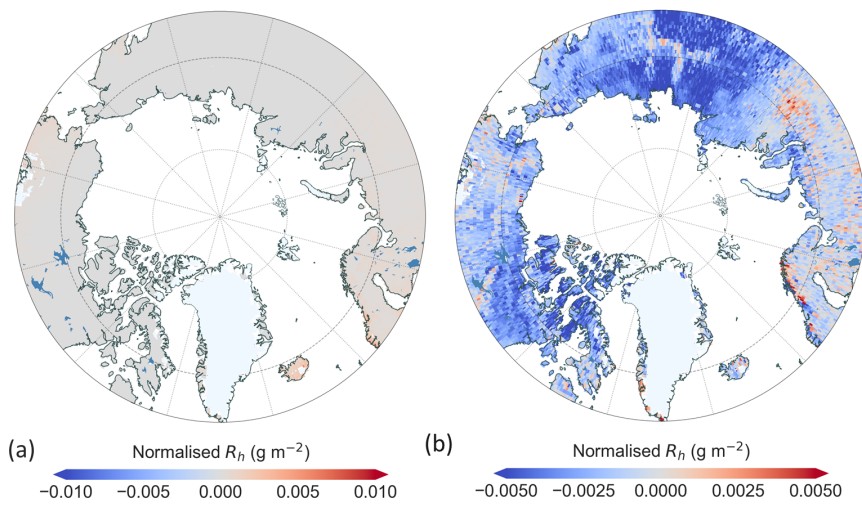

**Figure 8.** Differences between simulated *Dynamic* and *Static* winter (a) and summer (b) heterotrophic respiration normalised by soil carbon content. Differences are calculated by subtracting the *Static* from *Dynamic* simulation outputs.

We note an impact of the different snow schemes on summer productivity, caused by the different soil thermodynamics during
the spring and early summer period, as shown in supplement Fig. S3 (b). During summertime, the *Dynamic* scheme simulates
a larger uptake of carbon. The difference in net primary productivity (NPP) between simulations with the two snow schemes
for both winter and summer is shown in Fig. 9 (a) and (b), where positive NPP means more carbon uptake by the vegetation.
This figure shows some interesting contrasts: Winter NPP (a) is noticeably lower with the *Dynamic* scheme outside of the
permafrost region, related to increased autotrophic respiration, but higher in summer. Within the permafrost region, summer
NPP was lower with the *Dynamic* scheme while the winter showed no noticeable difference.

The simulated Pan-Arctic NEE is showed in Fig. 10 (a) and (b), where negative NEE values indicate a stronger uptake of
carbon by the ecosystem. The positive difference in winter NEE (a) shows that there is a higher carbon release in the winter
season for the *Dynamic* scheme in Central-Europe, Western-Russia and Scandinavia. The mean winter NEE of the *Dynamic*
scheme increased by 125 %. compared to the *Static* scheme, which relates to both the change in soil respiration and NPP.



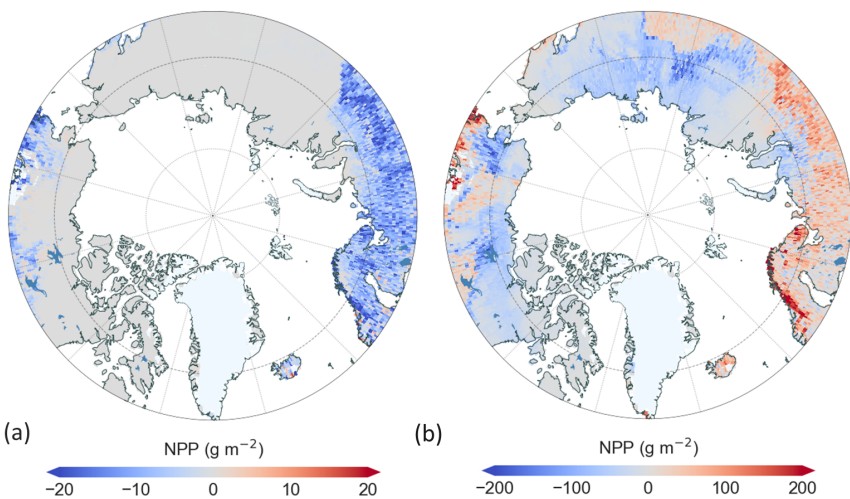

**Figure 9.** Differences between simulated *Dynamic* and *Static* winter (a) and summer (b) NPP. Differences are calculated by subtracting the *Static* from *Dynamic* simulation outputs.

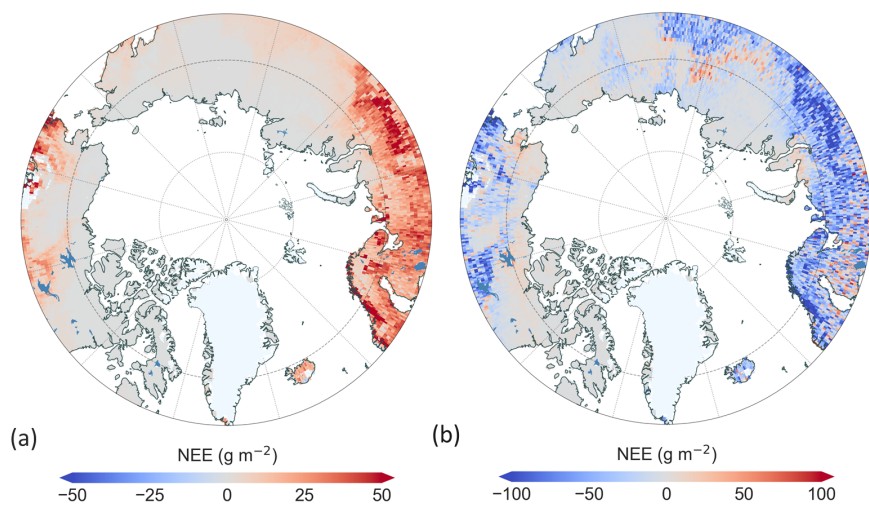

**Figure 10.** Differences between simulated *Dynamic* and *Static* winter (a) and summer (b) NEE. Differences are calculated by subtracting the *Static* from *Dynamic* simulation outputs. A positive difference (red) in NEE shows a higher carbon release using the *Dynamic* scheme. A negative difference (blue) in NEE shows that there is a higher release of carbon with the *Static scheme*.

### 3.3.3 Vegetation composition and distribution

Vegetation composition and distribution depend on the changes in physical and biochemical variables described in the previous section. Therefore, we investigated how the application of the two snow schemes affected vegetation distribution to determine



if there are shifts in dominant plant functional types (PFTs) as a result of using different snow schemes. The dominant PFT for
each simulated gridcell was determined by selecting the PFT with highest maximum LAI during the simulation years (1990-
2015). Using the *Dynamic* snow scheme, 22 % of the sites are dominated by summergreen low shrubs and boreal needle-leaved
evergreen trees (LSS and BNE PFTs, see Table A1). Prostrate dwarf shrubs, (SPDS), graminoid and forb tundra (GRT) and
boreal needle-leaved summergreen trees (BNS) accounted for 17, 10 and 10 % dominance. For an easier comparison between
the *Static* and *Dynamic* simulations, PFT classes were grouped into forest, open grass, shrubs and no vegetation categories
after determining the dominant PFT in each gridcell (classification based on Wolf et al. (2008)). This classification showed that
forest and grassland classes dominate to an equal extent, approximately. Shrubs prevail at 26 % of the simulation sites. There
is a negligible number of sites with no vegetation.

Changes in group dominance between the *Static* and *Dynamic* simulations occurred at 15 % of the sites, see Fig. S12 (a)
in the supplement. The Sankey diagram shows the direction of change between the three groups. Shrub domination mainly
transferred to forest-, and grass to shrub domination. This tendency is in line with the increased forest height and shrub cover
described by Myers-Smith et al. (2011), where increased temperature leads to a shift towards shrubs and trees.

The simulated vegetation groups for both applied snow schemes are plotted in Fig. 11 (a) and (b) for *Static* and *Dynamic*
schemes respectively. Sites with shifts in group dominance are scattered across the Arctic, but there are certain hotspots (see
also Fig. S12 (b) in the supplement). One such hotspot, a transition from forest to shrubs, can be observed in Eastern Russia
with the *Dynamic* scheme. A comparison with the biome mapping by Dinerstein et al. (2017) in Fig. 11 (c) shows that this forest
patch is in accordance with their tundra-taiga boundary. This suggests that the *Dynamic* scheme has an improved tundra-taiga
vegetation distribution.

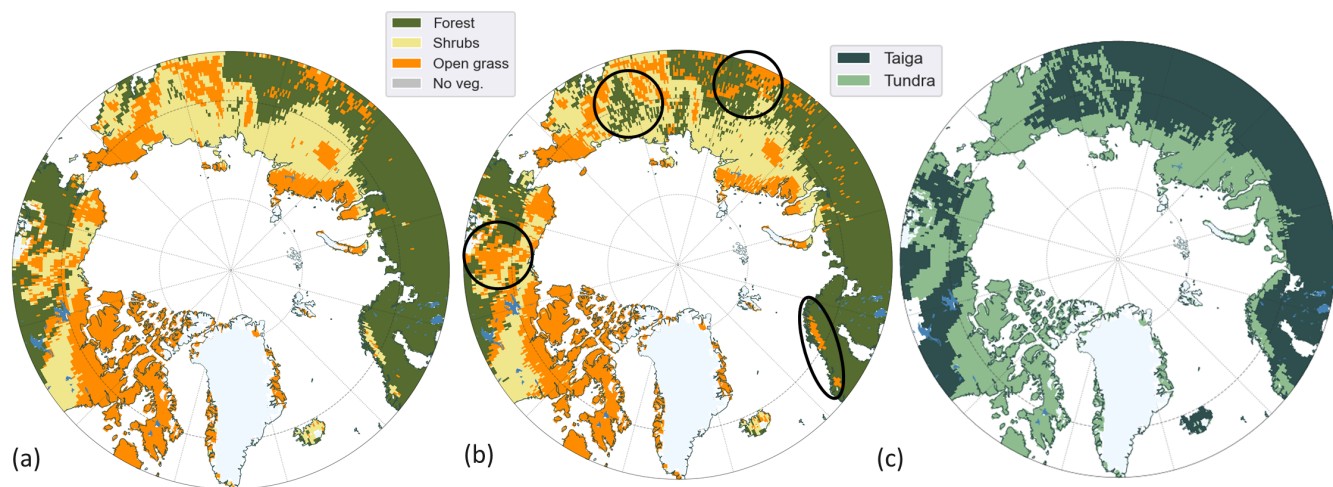

**Figure 11.** Vegetation distribution for the *Static* (a) and *Dynamic* schemes (b) and biome distribution (c) based on Dinerstein et al. (2017).
The main differences in vegetation distribution between the simulations are marked on (b).





**3.4  Cause-and-effect relationships**

To summarise the findings regarding the physical and biogeochemical processes, we created a flow chart showing observed
changes in modelled state variables and their connections. Fig. 12 shows the difference between simulations using the two
snow schemes – calculated by subtracting the *Static* from *Dynamic* results. Red box colours show that the *Dynamic* scheme
had higher values, and blue colours show that the *Dynamic* scheme simulated lower values than the *Static* scheme. The light-
ness and darkness of the colours indicates the magnitude of changes between the winter and summer seasons qualitatively.

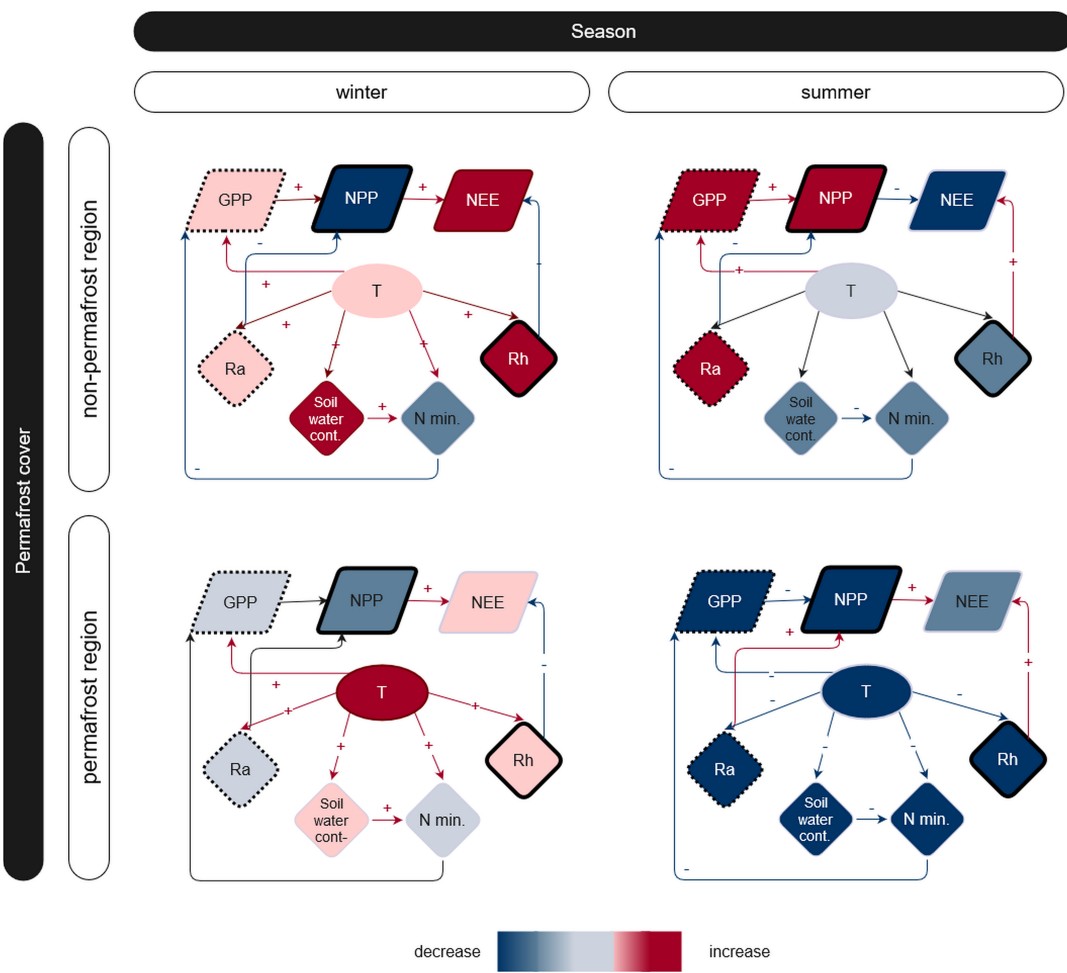

**Figure 12.** Relationship between variables during the winter and summer seasons, within and outside of the permafrost region. The colour
of boxes indicates the direction and qualitative magnitude of changes in the variables, based on the difference between *Dynamic* and *Static*
schemes. The colour of the arrows show whether there has been a net increase or decrease in the particular variable. Variables $GPP$ and $R_a$,
$NPP$ and $R_h$ are indicated by their matching borders, are then used to calculate the derived indices $NPP$ and $NEE$, respectively.



Table S3 in the supplement summarise the mean changes in these key variables. Considering the spatial pattern across the Arctic, we conclude that the pattern of changes and differences between the *Static* and *Dynamic* simulations vary depending on the presence or absence of permafrost cover. For a more detailed evaluation, process relationships are therefore divided into

permafrost and non-permafrost regions. Since snow depth only affects these variables indirectly, through insulation, it was not included in the feedback graph. The choice of snow scheme induced changes in near surface temperature ($T$), which is the key governing factor over these variables. In general, higher soil temperatures during the winter season prompt a positive response in respiration, soil water content and vegetation primary productivity. Soil temperature increased to a greater degree in the non-permafrost region during the winter season. This increase is observed for heterotrophic ($R_h$) and autotrophic respiration

($R_a$), soil water content and NEE. Nitrogen mineralisation decreased in the wintertime, with a larger decrease outside of the permafrost region. In contrast, summer months' average soil temperature showed an overall decrease in the permafrost region. This is due to the different thermal soil and snow dynamics in the two applied snow schemes. We observed a strong heat transfer using the *Static* scheme, resulting in insufficient insulation during the snow season. The same feature causes soil temperature to rise rapidly in the spring, when air temperature is already above zero but a snow cover remains present. Faster soil warming

leads to increased soil water availability that affects productivity. Respiration and NEE are slightly reduced for both permafrost and non-permafrost regions in the *Dynamic* scheme. Differences noted for the summer are smaller for all variables than in the winter season.

## 4 Discussion

### 4.1 Snow scheme dynamics

The site-level analysis shows that the new *Dynamic* scheme is able to simulate snow height and density adequately due to the implementation of physical processes and a dynamic representation of snow properties. The integrated mechanistic compaction scheme and phase changes within snow layers make it possible to simulate heterogeneous snow density and thermal properties within the snowpack. By altering snowpack structure, this influences the simulated snow density directly. Density regulates heat transport rate through snow layers by affecting thermal conductivity (Eq. 7): Lower density results in a more insulating

cover, whereas higher density and compacted layers are a better heat transferring medium and exhibit lower insulation. In the *Static* scheme, snow density was assumed constant through the snow season and across all study sites. Such static snow representation is unsatisfactory when simulating arctic conditions (Krinner et al., 2018). The new snow scheme provides an improved framework for a mechanistic snow season simulation.

The single site simulations (Sect. 3.1) provide reasonably consistent evidence that the new snow scheme's implementation leads to significant changes in near-surface temperature simulation – especially at Abisko, Bayelva and Samoylov. As shown by Chadburn et al. (2017), the site-wise model-data comparison is challenging since point measurements may not be representative of a larger area due to the complexity in topography and vegetation conditions. The model-observational fit may be





improved by using site-specific climatic forcing instead of a global gridded dataset.


To avoid site-specific problems in the interpretation of simulations, we also evaluated the model at a regional scale. By comparing the results of the Russian site simulations (Sect. 3.2) with those of Wang et al. (2016), we conclude that the development of the representation of snow in LPJ-GUESS significantly improved air-soil temperature and snow depth-soil temperature relationships. The *Dynamic* snow scheme's insulation capacity follows a quasi asymptotic trend, increasing with snow depth and

slightly levelling out after reaching the so-called effective depth at 30-40 cm (Slater et al., 2017). The insulation capacity is, in general, slightly lower than observations, with a notable underestimation when snow depth is below 20 cm. Nonetheless, these results are a vast improvement over the old *Static* scheme, as shown in Fig. (4).

The RMSE values (Table S2, supplement) also show that the *Dynamic* scheme better captures the observed soil temperature and snow depth relationship than the *Static* scheme. RMSE is slightly higher for the coldest air temperature regime for both

snow schemes. We note that the *Static* schemes' performance differs from what was shown in Wang et al. (2016). The reasons for these differences are developments of the model since then, in components other than the snow scheme, and the different meteorological forcing used in this study. Our results indicate that the enhancement of snow-related processes improved the simulation of soil thermodynamics in LPJ-GUESS and that the model can be applied for in-depth analyses across the Arctic.

### 4.2 Impact on physical and biogeochemical variables

The changes to the snow insulation capacity in the *Dynamic* scheme had a significant effect on permafrost conditions. Our pan-Arctic results showed that the *Static* scheme simulated near-surface soil temperatures that were too cold in winter and too warm in summer. Permafrost extent simulated with the *Dynamic* scheme agreed more closely with the permafrost estimate by Obu et al. (2019), as shown in Fig. 5. Comparison of these findings with other studies where a new snow scheme was introduced confirms that the model representation of snow strongly affects soil thermodynamics (Gouttevin et al., 2012; Wang

et al., 2013). Reliable soil temperature simulations are essential to study the permafrost-climate feedback. Biskaborn et al. (2019) concluded that recent warming trends of permafrost soils are partly due to an increase in snow insulation, accelerating its degradation. Both field observations and modelling studies have identified this close link between snow and permafrost conditions (Johansson et al., 2013; McGuire et al., 2016; Lawrence and Slater, 2010a). Identifying changes in permafrost underlain areas is important because of the potential increase in organic matter decomposition and release of greenhouse gases.

These aspects will be further evaluated in LPJ-GUESS with the new snow scheme.

We observed a general decrease in mean NPP during the winter and summer seasons. Considering the presence of permafrost, we noted an increase in GPP and NPP for non-permafrost underlain areas in summer. The significantly warmer winter soil conditions for the *Dynamic* scheme caused an increase in heterotrophic respiration due to faster litter decomposition rates and

increased microbial activity. Accordingly, soil respiration increased during the winter in the non-permafrost region. During the summer, there is an overall minor decrease in soil respiration, which can be attributed to the lower soil temperatures simulated


by the *Dynamic* scheme. The net effect of the above-discussed processes is an overall increase in carbon emissions during the winter and an increased uptake during the summer.

The impact of the new snow scheme on summer conditions was surprising. These differences were caused by the changes in spring snow and soil thermodynamics. During springtime, soils with the *Static* scheme warm more quickly, due to the lower insulation, which leads to an earlier thaw and increased soil water availability. The *Dynamic* scheme simulates a more realistic atmosphere-snow-soil heat transfer, leading to a slower temperature transition. The difference between the schemes diminishes towards the end of the summer. Overall, the simulated pan-arctic carbon fluxes are systematically lower than other published values (Efren et al., 2019; Rawlins et al., 2015; McGuire et al., 2012).

Besides the carbon fluxes, we also evaluated the simulated soil and vegetation carbon pools. Vegetation pools were not significantly different when applying the *Static* and *Dynamic* schemes, while clear differences were apparent for soil carbon. We found that the soil carbon pool is lower within the permafrost region and higher outside of the permafrost region. These results align well with the sensitivity study by Gouttevin et al. (2012), that highlighted that decreased soil carbon stocks can be attributed to a higher respiration rate and increased microbial decomposition rates.

The mean simulated soil carbon content was around 10 $kgCm^{-2}$, which is much lower than the 50-100 $kg\ C\ m^{-2}$ literature suggested range (Hugelius et al., 2014; Hugelius, 2012). The spatial pattern, however, is well captured by our simulations. This difference is most likely due to the fact that we only simulated grid cells with upland soils, while peatlands were not represented. The inclusion of peatlands would have led to a larger amount of soil carbon, since these ecosystems are characterised by waterlogged soils in which decomposition is suppressed – although carbon can be released as methane. Also, all organic matter is considered to be in the top 0.5 m of the soil in the current version of LPJ-GUESS, and is therefore only affected by the average soil temperature and moisture conditions down to 0.5 m, but not by conditions further down. These aspects will be taken into account in ongoing model development. Our analysis highlights that the observed differences between the *Static* and *Dynamic* schemes correlate well with the spatial pattern of near-surface soil temperature changes. The shortcomings in soil carbon simulation will be addressed and improved in the future, which will enable a more reliable carbon pool assessment.

### 4.3 Impact on vegetation dynamics

Satellite-based studies have identified an overall greening trend across the Arctic, in response to a warming from the 1980s until now. However, they also showed that this greening trend is not uniform and certain areas have actually experienced browning (loss of greenness) during this period (Berner et al., 2020; Myers-Smith et al., 2019). This may be partly due to damage to vegetation following extreme winter events Phoenix and Bjerke (2016). At the site level, a recent study by Niittynen et al. (2020) showed that winter thermal conditions are a strong control on vegetation patterns in arctic landscapes. Still, it is challenging to fully understand vegetation responses to warming solely from remotely sensed data or field observations, due to the scale dependency of interpreting trends in vegetation dynamics. Moreover, most field sites are highly concentrated in northern Scandinavia and Alaska, which leaves the full heterogeneity of the arctic and its ecosystems vastly under-sampled (Metcalfe





et al., 2018). With ecosystem models, we can fill in spatial gaps, identify feedback loops, and assess potential future changes.

Following the assessment of the new snow scheme's impact on biogeochemical variables, we compiled the simulated vegetation conditions with the two snow schemes. Figure 11 shows that the *Dynamic* scheme better captures the tundra and taiga
ecoregions, e.g. simulating a previously missing forest patch in Eastern Siberia. We found that PFT domination changed using the updated snow scheme in some gridcells. The main directions of change are from shrubs to forest and from grass to shrubs, indicating a tendency toward increased vegetation height – shown in Fig. S12 (a). Still, the forest-shrub border did not shift much in most areas. This finding, together with the minor difference in the vegetation carbon pool, suggests that there is no large change in biomass. A direct effect of snow-related changes on shifts in PFT dominance between the *Dynamic* and
*Static* schemes could be identified at a number of locations. These comparisons show that by changing the snow scheme in LPJ-GUESS, vegetation distribution and composition have also been affected. This strengthens the hypothesis that the studied physical and biogeochemical variables may have a significant effect on the simulated vegetation composition.

## 4.4 Outlook

It is well established that the Arctic is highly susceptible to climate change, and the ongoing warming has significant con-
sequences for the arctic system – even if we implement the most strict mitigation measures (Bruhwiler et al., 2021; AMAP, 2017). One of these consequences is a change in snow conditions. In the near future, snow thickness will decrease, caused by air temperature and precipitation changes, inducing a decrease in snow-covered area in the region (AMAP, 2017; IPCC, 2014). Due to a later onset and earlier spring melt, the snow season is expected to shorten under a changing climate. Moreover, northern high latitudes are predicted to be rain-dominated in the future (Bintanja and Andry, 2017; Johansson et al., 2011).
These changes will strongly influence soil thermodynamics, and the observed and projected changes will have a significant impact on arctic ecosystems (Bruhwiler et al., 2021). To be able to provide robust projections of the future, we need to account for a multitude of interlinked processes and feedbacks. Some of the current key areas are to assess the relative sensitivity of plant productivity to climate change, the development of decomposition rates and their net effect on the carbon budget.

Besides an assessment of geophysical and biogeochemical processes, LPJ-GUESS can also be used to explore future vegetation
trends to assess whether favourable growing conditions will induce further greening or whether new stressors will prompt local to regional scale browning (loss of productivity) (Myers-Smith et al., 2020). Studying future vegetation trends across the arctic is important from a global perspective. A potential decrease in snow covered area may significantly decrease surface albedo, which would enhance arctic warming. Consequently, changes in snow dynamics on a local scale influence carbon fluxes by altering soil thermal conditions and vegetation habitat. Evaluating snow-soil-vegetation feedbacks in future studies is therefore
relevant to further investigate climate change impacts on the Arctic in global scale land surface modelling.





## 5 Conclusions

This study shows that the representation of snow dynamics in a dynamic vegetation model significantly influences the simulated soil thermodynamics and related biogeochemical variables. We show, due to the improved snow insulation capacity, that the new *Dynamic* snow scheme simulates more realistic soil thermodynamics and permafrost extent than the old *Static* scheme.

The improved simulation of permafrost cover can be attributed to significantly warmer winter soil temperatures, which compare well to observations across 256 locations in Russia. We further showed the importance of an accurate snow scheme for the simulation of biogeochemical processes. Our results show that the intermediate complexity snow scheme had a significant impact on carbon fluxes. Heterotrophic respiration increased during the winter, which led to an increased carbon release during the cold season. We also identified differences in soil carbon content between the *Static* and *Dynamic* simulations. Although

the modelled soil carbon content was lower than literature values, the spatial pattern of low and high soil carbon content aligns well with observations. A differentiation between the seasons and accounting for permafrost presence highlighted the differences between the two sets of simulations. Wintertime carbon emissions were higher using the *Dynamic* scheme, both within and outside of the permafrost region. The differences between the simulations were larger within permafrost underlain areas. Besides spatial patterns, we explored seasonal differences, which showed that summertime conditions were also affected by

the representation of snow. In contrast to warmer soils in winters, soils were cooler in summer using the *Dynamic* scheme – especially in permafrost underlain areas – due to a delayed response to snow melt. These differences between the old and new snow schemes underline the importance of further developing winter processes as they may significantly affect the annual carbon budget.

These findings contribute to our understanding of the impact of wintertime changes on the arctic carbon cycle. We show that an accurate, dynamic snow scheme is essential to investigate the full complexity of snow-soil-vegetation relationships. Models are valuable tools to aid our understanding of large scale climate change impacts due to the sparse availability of observations in the Arctic. Addressing identified knowledge gaps in models is imperative to decrease the uncertainty around carbon balance estimates. Due to the large spread of observed and modelled seasonal and inter-annual cycle of carbon fluxes, it is not yet

possible to determine with high certainty whether the Arctic will act as a carbon source or sink in the future. To decrease uncertainty in simulations, contemporary modelling efforts are directed, on the one hand, at model inclusion (account for key, still missing processes) and, on the other hand, at refining process formulations using observational data (McGuire et al., 2012; Fisher et al., 2014, 2018).

In this study, we aimed to improve the representation of cold season process using non-growing season observations and find-

ings. This enhances the versatility and applicability of LPJ-GUESS as a tool to address the remaining uncertainties regarding climate change impacts at northern high latitudes and its consequences on a global scale. With this model, we have the ability to investigate complex ecosystem interactions under changing environmental conditions at multiple scales, considering nitrogen cycling, permafrost processes (freeze-thaw cycles, hydrology), stochastic vegetation dynamics and also the potential land cover and land use changes. Realistic soil temperature simulations are the first step to improve the simulation of greenhouse





gas emissions under different climate scenarios across the Arctic (Natali et al., 2019).

Our results show that by improving a process that appears only relevant in winter, such as snow, we not only decrease the uncertainty regarding physical and biogeochemical parameters during the cold season, but also improve simulations of soil conditions and the carbon cycle in the growing season. Further developments will aim at improving soil carbon content sim-

ulations and to better assess plant responses to future environmental conditions during the cold season. By accounting for snow-soil-vegetation interactions in all seasons of the year, we ensure more reliable projections of the future state of vegetation composition, permafrost stability and greenhouse gas exchange in a rapidly warming Arctic.

## Appendix A: Simulation details

**Table A1.** Applied PFTs and vegetation grouping, based on Wolf et al. (2008)

| PFT | Description | Vegetation group |
|-----|-------------|------------------|
| BNE | Boreal needleleaved evergreen tree | forest |
| BINE | Boreal needleleaved evergreen tree | forest |
| BNS | Boreal needleleaved summergreen tree | forest |
| IBS | Shade-intolerant broadleaved summergreen tree | forest |
| TeBS | Shade-tolerant temperate broadleaved summergreen tree | forest |
| HSE | Tall shrub, evergreen | shrub |
| HSS | Tall shrub, summergreen | shrub |
| LSE | Low shrub, evergreen | shrub |
| LSS | Low shrub, summergreen | shrub |
| GRT | Graminoid and forb tundra | grass |
| EPDS | Prostate dwarf shrubs (needleleaved, evergreen) | grass |
| SPDS | Prostate dwarf shrubs (broadleaved, summergreen) | grass |
| CLM | Cushion forb, lichen and moss tundra | grass |
| C3G | C3 grass | grass |





**Table A2.** Table with used variables, their description and units.

| Variable | Name | Value | Unit | Eq. number |
|---|---|---|---|---|
| $a$ | empirical variable | 109 | $kg m^{-3}$ | 3 |
| $b$ | empirical variable | 6 | $kg\ m^{-3}\ K^{-1}$ | 3 |
| $c$ | empirical variable | 26 | $g\ m^{-7/2}\ s^{-1/2}$ | 3 |
| $C_k$ | thermal heat capacity | | $J\ m^{-3}\ K^{-1}$ | 1,8 |
| $D_k$ | thermal diffusivity | | $m^2\ day^{-1}$ | 8,9 |
| $\eta_k$ | compactive viscosity factor | | $10^6 Pa\ s$ | 5 |
| g | gravitation | 9.81 | $m\ s^{-1}$ | 5 |
| $I_k$ | ice content | | $kg\ m_{-2}$ or $kg\ m^2$ | 4,6 |
| $k$ | layer index | | | 1,2,4,5,6,7,8 |
| $K_k$ | thermal heat conductivity | | $J\ m^{-1}\ K^{-1}\ s^{-1}$ | 7 |
| $k_s$ | empirical variable | 4000 | $K$ | 5 |
| $M_k$ | mass of overlaying snow layers | | $kg$ | 5 |
| $P$ | precipitation | | $mm$ | 2 |
| $\rho_k$ | layer density | | $kg\ m^{-3}$ | 1,4,5,6,7 |
| $\rho_t$ | reference snow density | 400 | $kg\ m^{-3}$ | 6 |
| $\rho_{fresh}$ | fresh snow density | | $kg\ m^{-3}$ | 3 |
| $\rho_0$ | reference snow density | 50 | $kg\ m^{-3}$ | 5,7 |
| $rw_{min}$ | empirical parameter | 0.03 | | 6 |
| $rw_{max}$ | empirical parameter | 0.1 | | 6 |
| $T_k$ | layer temperature | | $°C$ | 2,5,9 |
| $T_{max}$ | threshold for snow-water phase changes | 0 | $°C$ | 2,3,5 |
| $U_{10}$ | reference wind temperature at 10 m height | | $s^{-1}$ | 3 |
| $z_k$ | soil layer depth | | $m$ | 4,9 |
| $W_k$ | water content | | $mm$ | 6 |
| $W_{cap,max}$ | max. water holding capacity | | $mm$ | 6 |

*Author contributions.* AP and FJWP designed the research. Model developments were lead by AP and implemented by AP, DW and PM.
AP performed the model simulations and analysed the data. AP prepared the paper with contributions from all co-authors.

*Competing interests.* The authors declare that they have no conflict of interest.



*Acknowledgements.* We gratefully acknowledge support from the Swedish Research Council (WinterGap, registration no. 2017-05268) and the Research Council of Norway (WINTERPROOF, project no. 274711). DW has received funding from the European Union's Horizon 2020 research and innovation program under grant agreement No 641816 (CRESCENDO). DW and PM also acknowledges financial support
from the Strategic Research Area MERGE and the Swedish national strategic e-science research program eSSENCE.



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
