# Peer review of "Model simulations of arctic biogeochemistry and permafrost extent are highly sensitive to the implemented snow scheme in LPJ-GUESS"

_Biogeosciences, 2021_

## Author Comment (AC1)

**Response to the comment of Anonymous Referee #1**

We appreciate the time and effort from Reviewer #1 to provide detailed comments and suggestions on our paper. We address each comment below, where the reviewer's comments are shown in italics. The line numbers refer to the original document.

Notes on a recent model update:
We discovered a minor bug in the model after the submission of the initial manuscript, related to the handling of rain-on-snow events. After correcting this issue, we re-ran all the simulations and updated the figures and tables in this study. We conclude that the main findings presented stay unchanged, except for a lesser impact on vegetation patterns. As we moved multiple figures from the supplement to the main text following the suggestions of both reviewers, we decided to transfer the vegetation dynamics related figures to the supplement.

Summary of the changes to Figures and Tables (numbers refer to the original document):
We merged Fig. 8, 9 and 10 to one figure.
We moved Fig. 11 to the supplement.
Fig. 12 and its captions are updated.
We moved Fig. S2 to the main text.
We edited and moved Fig. S3 to the main text.
Fig. S11 was updated and included in the main text.
We updated and moved Table S3 to the main text.
We added a figure on soil carbon simulations to the supplement.

***General comments***
*The paper describes the implementation of a dynamic, multilayer snow scheme in the dynamic vegetation model LPJ-GUESS. The multilayer model's performance is compared to the single layer model's performance by evaluating the models' output with respect to observations of snow depth and soil temperature at the site level, soil temperatures and soil-air temperature difference on the regional scale, and dominant vegetation type distribution on the pan-arctic scale. The two models are also compared to each other with respect to soil temperatures and carbon cycle dynamics on a pan-arctic scale. The study therefore presents a thorough overview of the effects of switching to the new snow scheme, and in particular goes into detail on the causal relationships of the model's effect on carbon cycle dynamics. Interesting points are drawn out, such as the effect of snow in springtime on summer temperatures, and how the new scheme has an opposing effect on summer and winter soil temperatures. The study will be useful both for users of LPJGUESS, and those considering the impacts of dynamic snow schemes on model biogeochemistry.*
Thank you for the positive comments regarding the relevance of this study.

*While the paper shows that the implemented 'dynamic' snow scheme is a clear improvement over the 'static' snow scheme's configuration, the paper would benefit from and should include more discussion of what is responsible for the improvement in soil temperatures. In particular, comparison of the modelled snowpack conductivity and snow density between the two schemes should be included, as well as looking at the effect of changes in soil moisture content - an indirect effect of changing the snow scheme but a key control on soil conductivity and summer soil temperatures. Fortunately, some of this material is already in the supplement.*
Thank you for pointing this out. We revised, extended and re-arranged the relevant sections – see the detailed responses below.

*As previously mentioned, while cause and effect has been gone into in detail for the carbon cycle, similar detail has not been afforded to the changes in soil temperature. I would recommend that the discussion of snowpack dynamics in S2.1 be moved to the main text and expanded, as the effect of the model on snow density is perhaps the primary cause of the resulting changes to the modelled insulating effect of snow, and hence the ground thermal and biogeochemical changes, and there is currently very little quantitative discussion of the effect of the model on snow density and insulating factors.*

We did not conduct a model-data comparison regarding the internal snow variables (i.e. thermal properties) as it is challenging to obtain large scale datasets. Literature-sourced snow thermal conductivity values vary greatly and are in the range of 0.021-0.65 W m$^{-1}$ K$^{-1}$. The Dynamic scheme simulates thermal conductivities in the range 0.04 to 0.5 W m$^{-1}$ K$^{-1}$, with a mean value of 0.2 W m$^{-1}$ K$^{-1}$, which covers well the reference range. The constant thermal conductivity value used by the Static scheme (0.196) is a mean estimation that fits general conditions and matches the mean of the dynamic scheme quite well, but it does not capture the heterogeneity in Arctic snow conditions.

*Another primary control on soil temperatures is soil water content, particularly in the upper layers of the soil. Figure S11 shows that changing to the new scheme has an effect on July soil water content. Visually there is correlation between areas that have become drier in July as a result of the new scheme and those with reduced summer temperatures. This could be due to the soil being less conductive due to being drier, though there could also be the opposite causal relationship. In either case, I would recommend moving part of S11 into the main text and including some discussion of the relation between soil moisture and soil temperatures.*

Considering the comments of both reviewers we added a figure of soil moisture to the main text and discussed the soil moisture related findings in the relevant sections.

*Regarding the cause of the improvement in soil temperatures, it is not entirely clear that it is the dynamic and multilayer nature of the new scheme that is responsible. In figure three, for the static scheme, all sites seem to have reasonable snow depths but too low soil temperatures. This could indicate that one or more parameters in the original static scheme are set incorrectly, and that changing these would result in a similar level of improvement as changing the snow scheme. For example, the snow conductivity parameter could be set too high. While the investigation of adjusting these parameters would be a lot of work, comparing the total snowpack conductivity between the two schemes, perhaps on a new panel in figure 3, or alongside figure S3a would enable insight as to whether the original configuration is at least partly at fault.*

Fig. 4 as well as the study by Wang et al 2016 show that the *Static* scheme simulates a quasi-linear insulation curve. "Tuning" the constant snow related parameters would not result in better estimates of soil temperature that the *Dynamic* scheme provides since this quasi-linear behaviour would still be there. Furthermore, tuning would require variable fitting for each simulated gridcell, which is unrealistic in case of large scale simulations. Currently, the implemented *Dynamic* scheme is based on basic physical principles and is forced by climatic inputs that we find more fitting for our aims.

*Regarding the biogeochemistry, the effect of the new scheme on the modelled soil carbon is of course of key interest, and it would be nice to have more of an idea of how the model compares with observations. Line 395 says "The spatial pattern… [of soil carbon] is well captured by our simulations." This, however, is not backed up by comparison to observations. It is also not mentioned if this is an improvement over the original scheme. As mentioned in the text, comparison of the total soil carbon pool with observations is restricted by LPJ-GUESS not modelling peatlands and considering organic matter to be in the top 0.5 m of the soil. While not essential for this paper, it would be nice to compare the modelled and observed soil carbon for areas where the soil depth is only 0.5 m, or perhaps more*

*realistically compare the model to the carbon content in the upper 0.5m of the soil, which may be something for which you can find an estimate.*

We added a figure of absolute soil carbon content in the supplement to provide more information on the spatial pattern of the simulated soil carbon content. We plan to provide a more thorough model-observational data comparison, including carbon fluxes and pools in the future after the currently ongoing changes in the soil scheme has finished.

***Other points***

*Title and abstract – While your title reflects the key scientific application of the paper, I feel that a more appropriate summary would be 'The physical and biogeochemical effects of implementing a multilayer snow scheme in LPJ-GUESS', as your paper doesn't really aim to be a sensitivity study. At any rate, I would advise you to consider including the name of your model (LPJ-GUESS), and what you are specifically doing (introducing a multilayer snow scheme) in the title. If however the main aim is to answer specific scientific questions, these should be set up clearly in the introduction, and answered in the end. Currently, I think your abstract is a good summary of your paper, but it could do with some quantitative statements. Indeed, more quantitative statements would generally be welcome throughout the text when making comparisons.*

Thank you for these suggestions. Based on these, the title was revised, and we added "in LPJ-GUESS" at the end of the original title. We have also added more quantitative statements in the abstract to emphasise our findings.

*The implementation of the dynamic snow scheme is on the whole clear, however, in line 144 it says: "The computational cycle ends by rearranging the layers based on the depth thresholds, taking into account the potential liquid water content." More detail on how this is done is needed here to make your method reproducible. It would also be beneficial for readers unfamiliar to LPG-Guess to have a really brief overview (even a few sentences) on how soil thermodynamics and soil carbon are accounted for in the introduction to LPJGUESS at the start of section 2. For example, does the model have layers of soil, how deep is the soil, what permafrost processes are simulated?*

We have clarified this by adding: "First we re-calculate each snow layers' depth based on the amount of snow and liquid water using Eq. 4. We then re-arranged the layers by using the leaky bucket approach, where the snow layers are filled up from the bottom layer (closest to the surface). If the threshold depth is reached a new snow layer is initiated and the process continues until the total depth of the snowpack is distributed to the specific snow layers."

We also added more details in the Methods section regarding how the arctic processes are simulated by LPJ-GUESS. "Permafrost dynamics follows Wania et al. (2009a) and is simulated using the physical characteristics of 15, 10 cm deep soil layers. Soil thermodynamics is governed by climate and snow conditions, and the thermal properties of each soil layer depend on the ice, water, air, mineral and organic soil fractions. The layer-specific thermal properties define the rate of heat transfer through the soil column."

Wania, R., Ross, I., and Prentice, I. C. (2009), Integrating peatlands and permafrost into a dynamic global vegetation model: 1. Evaluation and sensitivity of physical land surface processes, Global Biogeochem. Cycles, 23, GB3014, doi:10.1029/2008GB003412.

*Section 5 - I'm not entirely sure as to the distinct purposes of each paragraph in the conclusions, maybe this could be made clearer?*

We revised the Conclusions section, and merged these paragraphs.

*Will you be including a code/data availability statement?*

We added a code and data availability statement.

*Plots are generally clear, however, colour bar scales should be altered for some plots to improve readability and to facilitate comparison. For example, in figure 8 it's very hard to see the distribution of winter respiration rate; maybe a different scale could be used? Also, the choice of two different scales for Figure 8a and Figure 8b make summer and winter difficult to compare, as well as making it appear as though there is more change in summer, while this might not be the case.*

All figures were revised and adjusted for better readability.

*Figure 12 is nice, however, there are a few things that could make it clearer. Firstly, to my eyes, the colorbar has a clear distinction between red (increase) and grey (neither increasing or decreasing), but a not-so-clear distinction between grey and blue (decrease). This results in it not being clear, for example, whether Ra for permafrost in winter is negative, neutral or even slightly positive but still very small. Maybe this could be made clearer with a separate colour in the middle, or a clearer changeover in the middle between red and blue. Also, the acronyms in this figure may not be immediately obvious to people who haven't read the text, so would be good to include in a key / the figure caption. Finally, I'm not sure I quite get the different box borders.*

This figure and its captions have been updated for improved readability – addressing the comments of both reviewers. Variables and their units are now shown in the figure caption.

*Figure 4 – The bottom right panel (delta T vs snow depth for the static scheme) seems wildly different to the observations and the dynamic scheme. Is the scale on the y axis wrong? On a separate note, it seems odd that there is so little variation in soil temperature with snow depth for the static scheme, so again I wonder if the static scheme has been set up with a poor value for the thermal conductivity and density. Again, it would be nice to compare these between schemes and go into why the new scheme has the effect it does.*

The axes were corrected in this figure.

Comparing the *Static* scheme's insulation curve in the Wang et al. 2016 paper and in Fig. 4 follows a linear trend, which is very different from the observed asymptotical pattern. This effective snow depth dynamics could not be achieved by simple changing the Static scheme's constant snow related variables. Additionally, the primary governing factor over the insulation dynamics is the snowpack thermal properties and not only snow depth. A more detailed description on how the new snow scheme influences physical and biogeochemical properties is now added to Section 3.1.

*Line 163 / Table S1 - Relatively little information is given in table S1 about these sites. I would say that including the coordinates of sites is essential. Climatic indicators would be nice. Site observations need to be referenced.*

Noted, the missing information is added to Table S1.

**Technical corrections**
*Line 18 - The sentence "These developments contribute to a better understanding of the Arctic's role in the global climate system" may be overstating things?*

This sentence has been revised. "These developments contribute to a more realistic simulation of arctic carbon-climate feedback."

*Line 28 – Since CO2 and methane is already being released, perhaps the sentence should talk about an increase in the release, or the change in the net warming effect?*

The Arctic currently acts as a sink (Bruhwiler et al. 2021, Virkkala et al. 2021). In this case we intended to emphasize the future potential of greenhouse gas emissions that might lead to changes in the Arctic carbon balance.

Virkkala, AM, Aalto, J, Rogers, BM, Tagesson, T, Treat, CC, Natali, SM, Watts, JD, Potter, S, Lehtonen, A, Mauritz, M, Schuur, EAG, Kochendorfer, J, Zona, D, Oechel, W, Kobayashi, H, Humphreys, E, Goeckede, M, Iwata, H, Lafleur, PM, Euskirchen, ES,

Bokhorst, S, Marushchak, M, Martikainen, PJ, Elberling, B, Voigt, C, Biasi, C, Sonnentag, O, Parmentier, FJW, Ueyama, M, Celis, G, St.Louis, VL, Emmerton, CA, Peichl, M, Chi, J, Järveoja, J, Nilsson, MB, Oberbauer, SF, Torn, MS, Park, SJ, Dolman, H, Mammarella, I, Chae, N, Poyatos, R, López-Blanco, E, Christensen, TR, Kwon, MJ, Sachs, T, Holl, D & Luoto, M 2021, 'Statistical upscaling of ecosystem CO2 fluxes across the terrestrial tundra and boreal domain: Regional patterns and uncertainties', Global Change Biology, vol. 27, no. 17, pp. 4040-4059. https://doi.org/10.1111/gcb.15659

Bruhwiler, L, Parmentier, FJW, Crill, P, Leonard, M & Palmer, PI 2021, 'The Arctic Carbon Cycle and Its Response to Changing Climate', Current Climate Change Reports, vol. 7, no. 1, pp. 14-34. https://doi.org/10.1007/s40641-020-00169-5

*Line 34 – It would be nice to quantify what "varies widely" means here.*

Range and reference has been added. "The predicted future carbon balance varies widely - between 641 Pg C loss to a 167 Pg C gain following RCP 8.5 (McGuire et al. 2018) - among models depending on the representation and level of detail of key processes …"

A. David McGuire, David M. Lawrence, Charles Koven, Joy S. Clein, Eleanor Burke, Guangsheng Chen, Elchin Jafarov, Andrew H. MacDougall, Sergey Marchenko, Dmitry Nicolsky, Shushi Peng, Annette Rinke, Philippe Ciais, Isabelle Gouttevin, Daniel J. Hayes, Duoying Ji, Gerhard Krinner, John C. Moore, Vladimir Romanovsky, Christina Schädel, Kevin Schaefer, Edward A. G. Schuur, Qianlai Zhuang. Proceedings of the National Academy of Sciences Apr 2018, 115 (15) 3882-3887; DOI: 10.1073/pnas.1719903115

*Line 54 – Maybe specify what these "key physical processes" are here.*

The key physical processes are now listed in this sentence.

L54. "Such schemes may not capture fine-scale internal snowpack processes such as the evolution of high-density wind slab layers, but they are complex enough to simulate key physical processes - compaction, freeze-thaw cycles and liquid water retention - that influence the thermal dampening property of snow."

*Line 59 – (Style) Maybe simplify to 'improve LPJ-GUESS's simulation of the insulating effect of snow'.*

The suggestion was noted and we revised this sentence.

*Line 61 – (Style) Change 'this' to 'the'*

Corrected.

*Line 64 – (Style) You switch to future tense here.*

We reviewed this sentence, but we could not spot future tense in line 64.

*Line 65 – I'm not sure what you mean by "We address the changes… ".*

We changed "address" to "analyse" for clarity.

*Line 73 – (Style) ", about 60° latitude," would be more suited to being in brackets rather than a double comma.*

Changed to brackets.

*Equation 2 – It might be nice to mention what precipitation is doing in this equation.*

We change the description in this section to introduce the P (precipitation) variable. P governs the rate of snow melt (along with air temperature) following an empirical method as defined in Eq. 9, by Choudhury and DiGirolamo, 1998. According to this equation, snow melts quicker if it's raining.

*Line 94 – It might be nice to mention the limitations of this assumption.*

We added a sentence in line 94 to re-iterate that the limitations of the *Static* snow scheme are the main features this paper is addressing.

*Line 99 – How are parameters a, b and c determined?*

Variables a, b and c in Eq. 3 are empirical parameters defined by Vionnet et al 2012. (doi:10.5194/gmd-5-773-2012). The reference is now added to the text where this equation is introduced.

*Line 103 – Is this minimum density often reached? Also, what is the physical mechanism which would impose this limit in reality?*

No, this condition occurs rarely, depending on the site conditions. According to literature fresh snow density is usually in the 30-300kg/m3 range. In theory, fresh density values below 100 kg/m3 may be possible when air T is around -20, -30 C deg., however these low snow densities are not common in the study regions (based on site observations from e.g. Sodankylä (Finland) or Zackenberg (Greenland)). Fresh snow density in the model is only affected by air T and wind, not accounting for e.g. the water content of fresh snow or small scale snow structure that could result in a more realistic fresh snow density simulation.

*Line 109 – Specify which physical properties are combined with the soil layer properties in this case.*

Note and added to the text.

*Line 110 – (Style) "In case all five layers are exceed…" should be 'In the case where all five layers exceed…"*

Corrected.

*Equation 5 – I'm not sure if $\rho_0$ has been defined yet.*

$\rho_0$ is now defined in the text with its constant value in brackets.

*Line 144 – Is a daily timestep used then?*

Yes, the snow cycle follows a daily time step

*Line 165 – Are overlapping time periods used?*

Our aim was to compare a general seasonal cycle of snow depth and soil temperature, therefore we used all the available years from the observational time-series presented in Table S1.

*Line 169 – How many Russian sites are there?*

We used 256 Russian sites in total (see line 171)

*Line 178 – "the influence of snow" -> 'the effect of changing the snow scheme'*

Corrected.

*Lines 187-190 – You are making the point the improvement in soil temperatures does not seem to be to do with improvements in snow depth. The logical conclusion is that it is the properties of the snow which have changed. This should be mentioned here. An analysis of where the change lies naturally follows this figure and should be discussed. While you may not have the level of detailed observations as in figure S2 for all sites, a new figure or panel similar to figure 3 comparing the difference in average thermal conductivity (and possibly density) of the snowpack between schemes, would allow this essential analysis.*

Figure S2 has been moved to the main text, showing the difference of the internal structure. We added a section describing the differences in internal snowpack structure and properties.

*Lines 192/3 says that "there is a smaller variance of modelled values of snow depth and soil temperature using the Dynamic snow scheme". This is true for soil temperatures, but not really the case for snow depths.*

This sentence is revised.

*On a separate note, would it be helpful to have the statistics covering just the winter period? The lower boundaries of the statistics may become more useful if this were the case.*

We updated Fig. 3 and its caption, the site statistics panel is recalculated, now excluding the summer months.

*Figure 3 – see point about snow densities/conductivity in the static scheme, also, is the soil wet enough for the static scheme in Abisko?*

To give more information about the differences in internal snow properties we included Figure S2 in the main text and expanded the description in the relevant sections (see response to previous comment).

*Line 203 – Still wondering how many sites there are in total.*

See answer in the simulation set-up section (n = 256), moved for clarity.

*Line 204 – "…has a better skill…" -> "has better skill"*

Corrected.

*Line 213 – "…has an improved skill to simulate…" - > "…has improved skill in simulating…"*

Corrected.

*Line 232 - Hypotheses need to be introduced, then tested. You have not introduced this as a hypothesis anywhere.*

The sentence has been revised, we removed the word "hypothesis" and simplified the sentence.

*Line 223 – This appears to read that implementing a compaction process in the model causes the snow depth to be greater. Are you sure that this is the case?*

This sentence was revised.

*Line 240 – much closer!*

The text was changed to "much closer".

*Line 247 – Figure S3a shows the temperature is increasing rapidly in the static scheme before snow melt, though you could possibly say this for the dynamic scheme, or for water availability rather than temperature.*

This sentence was revised.

*Lines 253/4 – The sentence "…the largest changes in snow depth, temperature and maximum ALD coincide' is only sometimes true, as can be seen from the figures. Temperature and ALD are of course directly linked but, again, most of the change in the insulative effect is not from snow depth change, and a large part of the summer temperature difference is from difference in moisture.*

This sentence was revised for clarity.

*Line 261 – exists -> exist*

This typo was corrected.

*Lines 262/3 – can you mention why changes in soil carbon sometimes do not coincide with soil temperature changes?*

Considering other comments, we revised Fig.12 and the related text where we describe more in detail the complex nature of the relationship between physical and biogeochemical variables.

*Line 264 – A small point: changes in soil temperature of course do influence soil carbon (you have assumed a rate of respiration change after all), so could this sentence be more specific?*

This sentence was adjusted.
L264. This shows that the changes in soil temperature influence the soil carbon content in the model. We can observe a contrasting difference in soil carbon content within and outside of the permafrost region, and in Fig. 12 we discuss how changes in soil conditions may affect soil carbon differently across the Arctic.

*Line 266 – Could you be more specific about how the normalisation is done?*

Details on the normalisation are now added.

*Figure 10 - Even though you are primarily comparing schemes, it would still be nice to show the absolute NEE from figure S9 in the main text so that the differences in the schemes can be compared to the absolute values i.e. would it be enough to flip the ecosystem from a source to a sink or is this a small effect. Maybe this could be talked about in the main text as well?*

We decided to move Table S3 to the main text and added a line with the annual NEE. We show that the effect of changing snow schemes is not enough to flip the Arctic sink to a source.

*Line 294 – "no vegetation" -> "mostly bare soil" (as we are comparing the dominant type of vegetation).*

This detail was changed in the text.

*Figure 12 - why is NEE decreasing for summer permafrost?*

Changes in NEE are related to the changes of Rh and NPP, as NEE is computed using these variables. During the summer Rh decreases more than NPP, see the complimentary numbers of mean changes now added to Fig 12.

*Line 320 – why does nitrogen mineralisation decrease in the winter? (this seems to be the opposite effect of what is expected from figure 12).*

We did not go into details regarding the potential drivers of nitrogen mineralisation as it was not the focus in our study. We provide the following potential explanation: the *Dynamic* scheme simulates higher vegetation carbon and therefore higher woody biomass. As a consequence, both soil carbon and woody litter increases. Woody litter having a higher C:N ratio induces immobilisation when decomposed. As a result of these processes, the Dynamic scheme simulates higher soil temperature and soil moisture content, woody litter availability, but lower net N mineralisation.

*Section 3.4 - This section could possibly so with a little more explanation to guide the reader through the cause-and-effect of what is quite a complex diagram.*

We added a short introduction to this section explaining that the complex, non-linear relationship and different feedback mechanisms between the studied variables makes it hard sometimes to track and grasp the direct effect of changes in the snow scheme. We added further explanation on how some of the studies variables are related and connected our statements to the updated Fig 12.

*Lines 322/3 – strong heat transfer results in insufficient insulation?*

We rephrased this sentence and changed to "rapid heat loss … resulting in insufficient insulation".

*Line 325 – maybe cross-reference figure S3.1 here?*

We decided to move Fig. S3 to the main text and discuss it more in detail in the text. Figure S3 was split into permafrost and non-permafrost panels considering the following comment.

*You could even split figure S3.1 into permafrost and non-permafrost panels.*

See the answer to the above comment.

*Line 333 - the sentence starting "By altering..." could be rephrased to make its relation to the preceding sentence clearer.*

We revised this sentence.

*Line 356 - delete the first comma in this sentence and change second comma to ', and also'*

This was corrected in the text.

*Line 356 – There's a bit of switching between present and past tense in this paragraph.*

We revised this paragraph.

*Line 358 – You've used the phrase 'improves the simulation of soil thermodynamics' (or similar) a few times now. It's a little picky, however, you haven't actually changed how the soil thermodynamics work, but you have improved the soil temperatures.*

Noted and changed to "soil temperature". We revised the text and changed the wording where we saw fit.

*Line 358 – "can be applied for in depth analysis across the Arctic" is a bit vague.*

This sentence was revised and extended.

*Line 364 – Does this need confirming?*

We have re-written this sentence.

*Lines 363 to 369 seem like they belong in the introduction.*

We tried to remind the reader what has been done in in this field previously and point out what is unique in this study. This section ties back to the information presented in the Introduction section.

*Line 371 – A 'however' at the start of the sentence 'Considering the presence of permafrost..' may help the logical flow with the preceding sentence.*

Thank you for the suggestion, we have rewritten this sentence including "however" for better readability.

*Lines 374 & 375 – I'm simplifying slightly, but this reads as 'respiration increases because respiration increases'?*

We revised this sentence to avoid confusion.

*Line 376 – 'can be attributed to' -> 'due to'*

Corrected.

*Line 377 - It would be nice to have a total for this, or at least some numbers! Maybe cross reference figure S3?*

We added Table S3 in the main text and provided mean annual numbers in the updated Fig 12 as well

*Line 384 – Could you give some numbers here?*
We provided the estimates of Virkkala et al. (2021) to motivate this statement.

*Line 386 - In most areas? Also, what does significantly different mean here? Maybe just compare the two, or say the total difference it makes in each case?*
This paragraph was revised for a more meaningful comparison and we avoided using "significant" to describe changes.

*Line 388 – "We found that the soil carbon pool is lower…" -> 'We found that using the dynamic model decreases the soil carbon pool...'*
This paragraph was revised.

*Line 421 – Hypotheses need to be introduced, then tested. You have not introduced this as a hypothesis anywhere.*
We removed the last sentence of the paragraph (line 421).

*Line 422 – is the word "significant" justified here?*
See the above comment and its response.

*Line 453 "and outside of" -> 'but especially outside of'*
Corrected.

*Line 453 "The differences between the simulations were larger within permafrost underlain areas." – This is true for the physical variables, but not the case for, say, NEE.*
Re-written the sentence.

*Line 465 – add reference.*
We added relevant references here.

*Line 478 - Apart from the tunga-taiga boundary & dominant veg types, I'm not sure you've done much comparison of the biogeochemical variables with observations, so can you say that you have decreased the uncertainty?*
The primary aim with this study was to show that changing the snow scheme in LPJ-GUESS not only had an effect on physical properties but also influenced the simulated biogeochemistry as well. The snow related physical processes are now improved and validated and we plan to provide a thorough comparison on model-observational carbon fluxes in the future – after the current model developments related to soil processes and non-growing season emissions are completed.

*Lines 476 to 482 perhaps have a bit of repetition.*
We consider these statements as the take home message from this study that worth highlighting at the end of the conclusions, therefore we prefer to leave the text as is.

***Supplement***
*S2.1 – Again, I feel like this should be in the main paper, as it's important to know that you can get the density right, and that's not really covered much elsewhere.*
We moved Fig. S2 to the main text and expanded the description on internal snow dynamics.

*S3.1 – It would be great to have an average thermal conductivity on here. Also, what area does this cover?*

This figure presents results from the Pan-Arctic simulations. We moved Fig. S3 to the main text and split the panels to permafrost/ non-permafrost regions. We added more discussion in the text on the changes in thermal conductivity using the *Dynamic* scheme. We decided not to add a thermal conductivity panel due to the lack of observational data and the trying not to overcrowd the already busy figure.

*Figure S9 - having got used to red = positive, blue = negative, it would be nice if this was also used for the Summer absolute NEE.*

We noted this comment and adjusted the figure accordingly.

*Figure S11 – Again, water content is a key control on summer soil temperatures - you can see the impact of the patches which are now drier on the summer soil temperatures - they don't warm up as much. Some comparison of this to changes in soil temperatures may be important in the main text.*

We considered the suggestions of both reviewers and added a figure on soil moisture content to the main text along with a discussion on the observed changes.

*S3.5 – "Fig. ??" - broken link?*

This typo was fixed.

*Figure S12a - It would be nice to show percentages for each of the pathways, especially since the fraction that remains unchanged is not shown. It might have been nicer to show the change in areal coverage (which would include the areal changes within sites of each PFT) rather than the change in the number of sites where that type dominates. However, it's probably not worth reanalysing for this plot.*

We noted this comment and now added percentages for the different pathways.

---

## Author Comment (AC2)

**Response to the comment of Anonymous Referee #2**

We thank Reviewer #2 for providing detailed, constructing comments the manuscript. We address each comment below, where the reviewer's comments are shown in italics. The line numbers refer to the original document.

Notes on a recent model update:
We discovered a minor bug in the model after the submission of the initial manuscript, related to the handling of rain-on-snow events. After correcting this issue, we re-ran all the simulations and updated the figures and tables in this study. We conclude that the main findings presented stay unchanged, except for a lesser impact on vegetation patterns. As we moved multiple figures from the supplement to the main text following the suggestions of both reviewers, we decided to transfer the vegetation dynamics related figures to the supplement.

Summary of the changes to Figures and Tables (numbers refer to the original document):
We merged Fig. 8, 9 and 10 to one figure.
We moved Fig. 11 to the supplement.
Fig. 12 and its captions are updated.
We moved Fig. S2 to the main text.
We edited and moved Fig. S3 to the main text.
Fig. S11 was updated and included in the main text.
We updated and moved Table S3 to the main text.
We added a figure on soil carbon simulations to the supplement.

***General comments***
*This article presents the new snow module implemented in LPJ-GUESS and shows how this newly implemented module affects the simulation of snow depth, soil temperature and ultimately biogeochemistry and vegetation distribution. The study is focussed on the arctic and sub-arctic regions of the northern hemisphere.*
*Showing how physical realism is important for biogeochemistry although the traditional separation between the two scientific communities tends to disappear.*
We are glad to see the relevance of this study acknowledged.

*This paper is well written in a clear style. I have however a few major comments, followed by detailed comments and questions.*
- *I understand the need to limit the length of the paper but the authors tend to describe differences in simulation results without explaining them, or the explanation given is not sufficient. The authors present results for snow depth, soil T and water content at 25 cm depth and C fluxes, but they don't clearly show how their change in snow model affects those variables. They show results for the winter and summer seasons but the explanation often has to do with what happens during the spring, and the reader doesn't have a figure about spring*

  We acknowledge the comments regarding the necessity to provide more details how snow structural changes influenced the studied physical and biogeochemical entities. To address this shortcoming, we moved Fig. S2 (Zackenberg site analysis) to the main text and added more details in Section 3.1 about how snow structure and physical properties changed and lead to the differences in soil temperature and biogeochemical processes. We edited Fig. S3, splitting the plot to represent permafrost and non-permafrost regions. This figure was moved to the main text to provide information on the seasonal pattern and relationship of snow depth, soil temperature, soil water content and carbon fluxes.

- *The authors present site level simulations that were forced by large-scale atmospheric forcing (CRU). I don't see the point since some of these sites, like Zackenberg, have very detailed meteorological data. It also makes me wonder if the authors did just 2 pan-arctic simulations and compared the results with local data or if they used local data (like soil texture, or else) to perform these site-level simulations.*

  We did not intend to calibrate the model to specific sites, but aimed at achieving an overall improved model-observational fit across the Arctic. The LPJ-GUESS model is most commonly used with the standard, gridded climatic forcing in studies, thus we found it relevant to use the global drivers in this study to provide information on the model's skill for future projects. All simulations in this study ran with CRU climatic input without using site specific data. This was done to be able to compare the model's behaviour on different scales using the same simulation set-up. (For reference, the simulations were set up similarly for the Russian regional simulations in the referenced Wang et al. (2016) study.)

- *I believe an analysis of the changes in water content might be necessary*

  Considering the comments of both reviewers, we included a figure on soil moisture in the main text and added a description on how the changes in the snow scheme influences soil moisture conditions.

- *There are quite a few errors in the figures, errors in the units of the equations*

  Figures, tables and equations have been revised and corrected where found necessary – see specific comments below.

**Detailed comments:**

*Eq 6 : as written, the units of this equation don't match: Ik is supposed to be in kg/m2 and rw don't have units (according to Table A2), so Wcap can't be in mm.*

Following general assumptions 1 kg of liquid water over 1 $m^2$ surface corresponds to 1 mm coverage, therefore these units could be applied in Eq. 6. We acknowledge that the used units can cause confusion and for the sake of an easier overview we changed the units of $W$ and $W_{cap}$ in Table A2 to $kg/m^2$.

*Figure 2: sublimation is not taken into account?*

No, sublimation is not taken into account in the *Dynamic* snow scheme. The only way snow layers (snowpack) lose volume (SWE, snow water equivalent) is through the melting process.

*as represented there is no snowfall on bare ground and I don't understand why the rain on bare ground affects the thermal properties. Are those boxes related to snow only or to snow and soil processes? If soil processes are represented, then why would thermal properties be based on density ? Is it an average of soil, water and ice density?*

This figure shows processes related to the snow cycle. We made some adjustments to the figure and updated its captions for a more straightforward understanding. We now refer to the pathways as "snowpack present" or "snowpack absent" instead of the previous wording.

*S1: typo : I don't think the Q10 was changed from 200.5 to 2.9 !*

It is not a typo, the adjustment was made using recently published literature suggestions (Natali at al 2019.)

Natali, S.M., Watts, J.D., Rogers, B.M. *et al.* Large loss of $CO_2$ in winter observed across the northern permafrost region. *Nat. Clim. Chang.* 9, 852–857 (2019). https://doi.org/10.1038/s41558-019-0592-8

*L190-191: as shown in Figure 3, soil temperature is higher than observations in summer only for 2 sites out of the 5: Kytalyk and Samoylov. Also, it is not really discussed in the following sections.*

The related and following sentences have been removed from the text.

*L192-193: I agree that soil temperatures have a smaller variance with the Dynamic than the static run but that is not true for snow depth, except in Abisko.*
We removed snow depth from this sentence, as a response to a comment from Reviewer 1.

*Figure 3 : are the site statistics calculated with monthly values ? or monthly anomalies ? (departure from the average seasonal cycle). Since the average seasonal cycle is shown, I would show the statistics from the anomalies.*
We updated this figure and excluded summer months from the statistical panel. We are showing the absolute numbers of average monthly values, and we prefer to keep the range of absolute values in this figure.

*Figure 4: The Y axis of the observed soil T and static snow results are mixed up: The observed soil T axis should be -40 to 0 like the simulations. Similarly, the Y axis from the snow depth for the static scheme should go from 0 to 35 like the other 2 snow panels.*
Thank you for pointing this out, this figure has been updated.

*L222 : Figure 5a doesn't show at all large changes in Scandinavia and in Western Russia. The blue in coastal Norway can be mistaken for the coastal line and lakes (like in Canada and Finland) and western Russia seems less red than central Siberia for instance and shows much less change than N-E Canada (Baffin etc) or far East Siberia. I suggest other colors (especially the blue that is impossible to distinguish from lakes) and may be give longitudes instead of "western Russia"*
All the figures were revised and lakes are now shown in white to avoid misunderstandings. The reason behind changes in the coastal region is the high amount of precipitation in the region and newly introduced compaction scheme affecting snow depth.
To address the second half of the comment, we reviewed and changed how we refer to the most affected regions: we write coastal Norway (instead of Scandinavia) and Western Siberia (instead of Russia).

*Figure 5c: the TTOP PF rectangle in the legend should be solid grey (not just the border)*
This detail is now corrected.

*L235: This explanation is wrong. Maximum ALD happens at the end of the summer, early fall. The authors can't use the warmer winter temperatures as explanation for the deeper ALD ! According to figure 6, Dynamic has cooler temperatures in summer, not warmer. I would assume that it is more related to the speed at which the soil refreezes in the fall. Static refreezes earlier and faster, hence stopping the summer melt earlier and reducing the increase in ALD.*
We see the point by the reviewer that this explanation is plausible but we suggest keeping our current discussion on the drivers of ALD changes. We acknowledge that there are multiple factors that can account for the changes in ALD and define the maximum ALD. We adjusted Section 3.4 to provide a more thorough explanation on the complex relationship between the potential drivers i.e. soil temperature, soil moisture, timing of snow melt etc. We can observe in the presented figures (e.g. Fig. S6 top row) that the largest changes in winter soil temperature occur at the edges of the permafrost region. Even though summers are cooler, we can see the direct winter impact on the mean annual soil temperatures that define the estimated permafrost extent (see Fig. 5 (c)).

*Section 3.3.2: in all this section (text and figures) and in the supplementary material, the units of the C fluxes are wrong. They should be in mass of carbon per unit area **per unit time.** The authors wrote g/m2 but the reader has to wonder if it is per year, per month, per season…*
Figures in this section show the mean spatial pattern of variables averaged over a season (winter or summer) as stated in section 3.1.1. All of the Pan-Arctic figures and related sections show average conditions over the winter and summer seasons for the period 1995-2015, if not stated otherwise. We adjusted figure captions to make this clear.

*L266: "soil carbon outputs were used to normalize Rh .."; the authors should say how here or in the legend of Figure 8 because I don't understand the units of the normalized Rh (g /m2)*

See the related comment of Reviewer 1. We now added a more thorough explanation on how the normalisation was done and included this information in the figure captions. In this case, the plotted $R_h$ represents the fraction of soil carbon respired per season (winter or summer). Considering this comment we reviewed the units and changed it to "fractions". We hope that these changes collectively can help to better understand Fig 8.

*L 268: "winter respiration … increased, except for …" : this is totally invisible on Figure 8a with the current colormap*

The color map of Fig 8 (and all other figures) has been revised and adjusted for better readability.

*L 270-280 : - the authors should refer to the figures in Annex Fig S7 to S9*

We noted this comment and made sure refer to the supplementary figures in an orderly manner in the main text.

*- Figure S8 winter (and Figure 9a) : There is something wrong here Static has values around 0.250 g/m2 (dark green) where Dynamic only has 0.125 but the difference doesn't show anything as if a mask had been applied. Also, white for negative NPPs is not very visible.*

We re-plotted this figure with a different, diverging color map for increased readability. This cleared up the confusing pattern caused by the use of inappropriate color map.

*L 275 : "increased autotrophic respiration" : why ? I know winter temperatures of the soil at 25 cm are higher. But how does that affect autotrophic respiration? I guess the question is how is vegetation temperature calculated ?*

This change can be attributed to higher soil temperature prompting increased root $R_a$. Other contributing factors involve more roots due to increased vegetation carbon and higher maintenance respiration.

Root respiration is calculated as function of the carbon biomass of roots, soil temperature, root C:N mass ratio:

$$root\ resp = \frac{\text{maintenance resp. coeff } \times \text{ scaling factor } \times \text{ root C mass}}{\text{root C: N mass ratio } \times \text{ respiration Q10 T response}}$$

*L275 "but higher in summer". Again why is that ? T is lower in summer. Could it be linked to water content ?*

We revised this paragraph and considering the received comments we now edited Fig. S3 and moved it to the main text (see response above). By doing this, we give a more detailed information on NPP, NEE, soil temperature and soil moisture relationship. During the summer, the Dynamic scheme simulates higher NPP in non-permafrost region and lower in permafrost underlain areas. Changes during the winter months are on a much smaller scale, where indeed we can observe a marginally lower NPP. The changes seen in NPP can be derived from the GPP and Ra. The balance of the changes in these variables govern the changes we note in NPP.

*L279-280 : when discussing NEE, it would be interesting to have annual values too.*

We added annual values for NEE in Table S3 that has been moved to the main text.

*Fig S9: the grey is really disturbing because it can be confused with light blue. Is Nee decreasing in large parts of Siberia and Canada in winter ?*

The figures' colour palette has been revised and changed for better readability. NEE is now presented using a diverging scale and unified scale for both winter and summer seasons.

*L 295 – text section S3.5: typo Fig ??*
This typo has been corrected.

*Figure 12: The signs on the figure are not explained. A typo may be ?*
*I don't understand the colors of the arrows – I think the caption is not completely correct : it states that "The colour of the arrows show whether there has been a net increase or decrease in the particular variable" but this is shown by the color of the box it-self. I assume the color of the arrow means how the change in one variable affects the change in the other (higher soil temperature favors higher heterotrophic respiration for instance). But that doesn't seem the case either because increasing Rh should favor increasing NEE, but the arrow is blue for the 2 left panels. I might have missed something but the figure should be better described.*
This figure and its captions has been updated for improved readability – addressing the comments of both reviewers. We listed the presented variables and their units and reviewed the connecting arrows as well.

*L 320: 3N mineralization decreased in wintertime": why is that ?*
See our answer to a previous comment above.

*L344 : "the model-observation fit may be improved by using site-specific climatic forcing" : yes indeed ! why not do that ?*
See the response to this comment in the general comments sections.

---

## Author Response (AR2)

**Final author's response on BG-2021-121**

Dear editor,
We appreciate the acceptance of our manuscript for publication. We reviewed the figures prior to the final upload and decided to make a minor adjustment to Fig. 12 to improve readability. (This adjustment only affects one of the presented variables and we noted this change in the figure caption.)

Kind regards,
The authors